



# A Global Classification Dataset of Daytime and Nighttime Marine Low-cloud Mesoscale Morphology Based on Deep Learning Methods

Yuanyuan Wu[1], Jihu Liu[2,3], Yannian Zhu[1,2,3,*], Yu Zhang[2], Yang Cao[2,3], Kang-En Huang[2], Boyang Zheng[2], Yichuan Wang[2], Yanyun Li[2], Quan Wang[2], Chen Zhou[2], Yuan Liang[4], Jianning Sun[2], Minghuai Wang[2,3], Daniel Rosenfeld[2,5]

[1]Nanjing-Helsinki Institute in Atmospheric and Earth System Sciences, Nanjing University, Nanjing, 210023, China
[2]School of Atmospheric Sciences, Nanjing University, Nanjing, 210023, China
[3]Joint International Research Laboratory of Atmospheric and Earth System Sciences & Institute for Climate and Global Change Research, Nanjing, 210023, China
[4]TianJi Weather Science and Technology Company, Beijing, 100000, China
[5]Institute of Earth Sciences, The Hebrew University of Jerusalem, Jerusalem 91904, Israel
*Correspondence to*: Yannian Zhu (yannian.zhu@nju.edu.cn)

**Abstract.** Marine low clouds tend to organize into larger mesoscale patterns with distinct morphological appearances over the ocean, referred to as mesoscale morphology. While prior studies have mainly examined the fundamental characteristics and shortwave radiative effects of these mesoscale morphologies, their behaviour in the nighttime marine boundary layer (MBL) remains underexplored due to limited observations. To address this, we created a global classification dataset of daytime and nighttime mesoscale morphologies of marine low clouds using a deep residual network model and Moderate Resolution Imaging Spectroradiometer (MODIS) infrared radiance data, with machine-learning-retrieved all-day cloud optical thickness aiding in model training. We analysed day-night contrasts in climatology, seasonal cycles, and cloud properties of different cloud morphology types in this study. Results show that relative frequency of occurrence (RFO) of closed mesoscale cellular convection (MCC) significantly increase at night, while that of suppressed cumulus (Cu) shows a remarkable decrease. Disorganized MCC and clustered Cu display a slight frequency increase during night. In addition, solid stratus and three MCC types exhibit distinct seasonal variations, whereas two cumuliform types show no clear seasonal cycle. Our dataset extends the study of mesoscale cloud morphologies from daytime to nighttime and $1^\circ \times 1^\circ$ resolution makes it better match with other climate datasets. It will provide a vital foundation for further research on the interactions between cloud morphology and climate processes. Our dataset is open-access and available at https://doi.org/10.5281/zenodo.13990646 (Wu et al., 2024).

## 1 Introduction

Marine low clouds cover the vast majority of the ocean and have a pronounced impact on the Earth's radiation budget. Their daytime shortwave cooling effect and night-time longwave warming effect are essential in modulating the climate variability (Klein and Hartmann, 1993; Eytan et al., 2020). These radiative effects are known to be sensitive to the cloud types due to their different cloud properties, such as cloud fraction and albedo. Traditional ground-based observations have historically





classified individual marine low clouds using the World Meteorological Organization (WMO) cloud types such as cumulus (Cu), stratocumulus (Sc) and stratus (St) (Zhang et al., 2018; Li et al., 2022; Guzel et al., 2024). However, satellite imagery

shows that these individual clouds tend to organize into larger mesoscale patterns with distinct morphological features that are not easily discernible from the limited perspective of ground observation instruments. These mesoscale cloud patterns, referred to as cloud mesoscale morphologies, have been shown to exert different radiative effects on climate (McCoy et al., 2017; McCoy et al., 2023; Mohrmann et al., 2021), and also reflect the intricate physical processes of the underlying marine boundary layer (MBL) (Wood, 2012; Bony et al., 2020; Eastman et al., 2022; Liu et al., 2024; Mohrmann et al., 2021).

Previous studies have identified several critical environmental factors that influence the evolution of marine low-cloud morphologies. In the mid-latitudes, open and closed mesoscale cellular convection (MCC) clouds are both affected by cloud-top longwave radiation cooling (Wood, 2012), but the surface fluxes dominate the open MCC when there is a strong cold advection such as polar outbreak. As a result, the passage of mid-latitude cold air outbreaks serve as key triggers for the transition from closed to open MCC (McCoy et al., 2017; Tornow et al., 2021). In the subtropics, precipitation promotes the

organization and sustainment of open cell structure and dominates the transformation of closed to open MCC clouds (Savic-Jovcic and Stevens, 2008; Feingold et al., 2010; Yamaguchi and Feingold, 2015; Eastman et al., 2022). In contrast, closed MCC tend to evolve into more disorganized cumulus under conditions of warmer sea surface temperature and increased entrainment of dry air at the cloud top (Eastman et al., 2022; McCoy et al., 2023). Apart from meteorological influences, aerosols can also initiate these transitions by modulating precipitation. High aerosol concentration suppresses the precipitation

and favors the maintenance of closed MCCs, while the scarcity of aerosols promotes the generation of widespread precipitation, leading to the conversion toward open MCCs (Stevens et al., 2005; Rosenfeld et al., 2006; Petters et al., 2006; Xue et al., 2008). With global warming and emission reductions, there is a high likelihood that meteorological factors and aerosols will change accordingly. This raises several important questions regarding low-cloud feedback, such as whether the mesoscale morphology of low clouds will change as the climate warms and how these changes will affect radiation.

An objective classification of mesoscale morphology from satellite observations is essential for facilitating a more systematic investigation of these questions. In recent years, deep learning methods, especially those based on convolutional neural networks (CNNs), have proven particularly effective in the objective classification of mesoscale cloud morphology in satellite images, enabling the subsequent generation of potentially informative marine low cloud datasets for further study. Wood and Hartmann (2006) trained a three-layer neural network to classify daytime cloud morphology in $256 \times 256$ pixel

scenes into four categories: no MCC, closed MCC, open MCC, and cellular but disorganized MCC. Their work was pioneering but limited to only subtropical regions for 2 months. Subsequently, Yuan et al. (2020) subdivided the cellular but disorganized category into disorganized MCC, clustered Cu and suppressed Cu, and developed a global dataset of these six cloud types using a fine-tuned VGG-16 model. Their dataset has higher spatial resolution, at $128 \times 128$ pixel, but also only includes classifications for daytime scenes. Watson-Parris et al. (2021) employed a pre-trained CNN model to detect pockets of open

cells (POCs) ($224 \times 224$ pixel) in three main marine stratocumulus regions during daytime. Moreover, Schulz et al. (2021)



trained an object detection model to identify four larger scale cloud morphologies in the trades of North Atlantic, categorizing them into types including "sugar", "gravel", "flowers", and "fish".

The datasets mentioned above have been utilized for various downstream tasks, such as quantifying shortwave cloud radiative effects and identifying key controlling factors of different cloud morphologies (Bony et al., 2020; Mohrmann et al.,
2021; Watson-Parris et al., 2021), quantifying shortwave cloud feedbacks resulting from changes in morphology (McCoy et al., 2023), and investigating aerosol-cloud interactions across different morphologies (Liu et al., 2024). However, most current studies focus on the role of morphology in daytime shortwave radiation, with a notable lack of understanding regarding longwave radiation, particularly nighttime longwave radiation, primarily due to the scarcity of nighttime observations of cloud morphologies. Although there are a few geostationary satellite-based studies that give a morphological classification during
night, they are also limited to regional scales and lack a global-scale classification dataset (Lang et al., 2022; Segal Rozenhaimer, 2023).

Datasets on nighttime cloud morphology are scarce, yet it is essential for investigating cloud-climate feedback. Closed MCC clouds have been shown to peak at night (Lang et al., 2022) and the subsequently increased cloud cover could lead to a rise in surface temperature by enhancing downward longwave radiation (Dai et al., 1999), which would further reduce the
diurnal temperature range and affect sea breeze-like circulations (Vose et al., 2005; Davy et al., 2017; Cox et al., 2020). Climate model suggests that, compared to daytime, the slower decline trend in night-time cloud cover could raise the global temperature and amplify climate warming (Luo et al., 2024). However, how these cloud morphology types behave under the influence of the nighttime MBL regime and how much they contribute to nighttime cloud cover variation remain unclear. In addition, marine precipitation is more frequent at night (Dai, 2001; Dai et al., 2007), with its intensity strongly depend on cloud
morphology types (Muhlbauer et al., 2014). Therefore, comparing the differences in cloud morphology between daytime and nighttime may help explain the uneven distribution of precipitation, as well as improve our understanding and prediction of global precipitation changes against the backdrop of climate warming.

Motivated by the aforementioned issues, a new $1^{\circ} \times 1^{\circ}$ classification dataset of daytime and nighttime marine low-cloud mesoscale morphology was generated in this study using a residual network model. In contrast to previous cloud classification
datasets, our dataset provides global coverage over a 5-year period (2018–2022) and can better integrate with other reanalysis data to offer more precise information about the meteorological conditions and environmental aerosols. Section 2 introduces the datasets and methods. Section 3 presents the training results and the contents of our dataset. The advantages and limitations of this dataset are discussed in Section 4. Section 5 states the data availability and Section 6 concludes.





## 2 Data and methods

### 2.1 Cloud type classifications

We adopted the classification scheme in Yuan et al. (2020) for mesoscale morphological classification of marine low clouds, and examples of each morphology classification are shown in Fig. 1. Solid stratus clouds are created by cloud-top radiative cooling and have a flat and uniform surface. Closed MCC clouds are stratocumulus driven by longwave radiative cooling and surface fluxes and display distinctive honeycomb-like structures with clear and descending edges. Open MCCs have a clear

descending region in the center, which is surrounded by several active shallow convective clouds. They appear in more unstable environment and typically generate heavier drizzle, lower shortwave reflectance, and greater transmissivity than closed MCC (Wang and Feingold, 2009; Muhlbauer et al., 2014). Disorganized MCC are a mix of convective elements and extensive stratiform clouds, marked by larger droplets and lower optical thickness. They tend to occur in a drier troposphere and over warmer oceans (Wyant et al., 1997; Bretherton et al., 2019). Clustered Cu refers to the aggregation of shallow, vigorous

convective elements, while suppressed Cu consists of individual, scattered cumulus clouds that occasionally form linear or branched patterns. Both of them are frequently and extensively observed over warm tropical oceans.

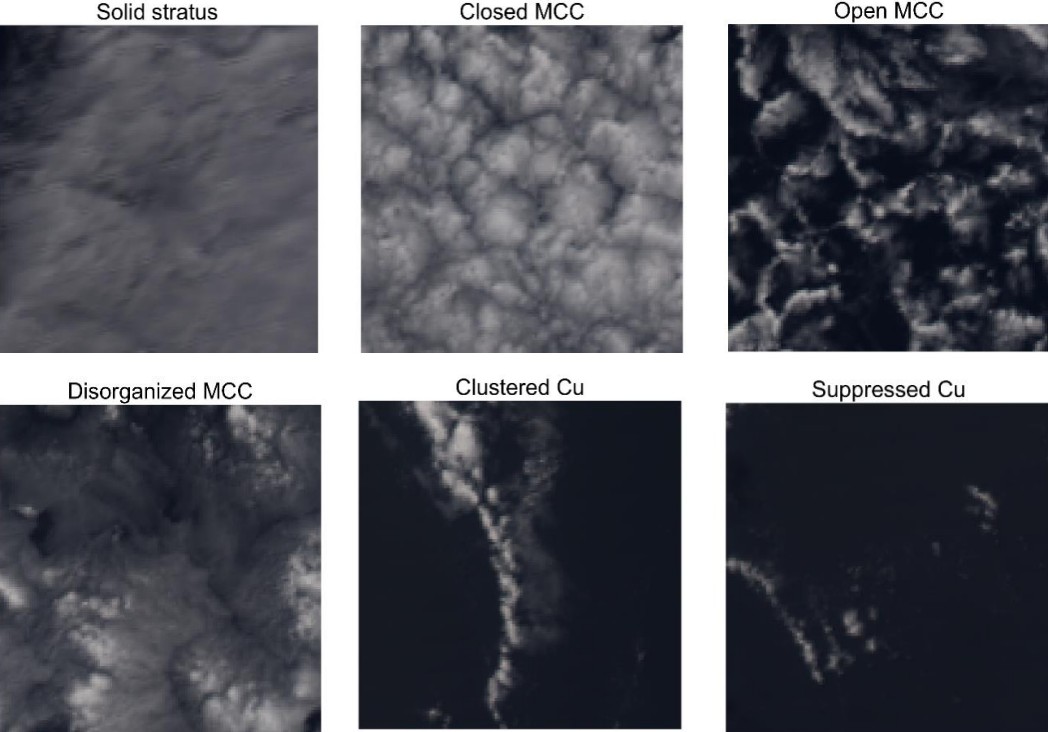

**Figure 1: Example scenes of six cloud morphological types: solid stratus, closed mesoscale cellular convection, open mesoscale**
**cellular convection, disorganized mesoscale cellular convection, clustered cumulus and suppressed cumulus. They are visible light images composed of channels 1, 4, 3 and the spatial resolution is $1^o \times 1^o$.**



## 2.2 Data

The primary observation data utilized in this study were derived from the Moderate Resolution Imaging Spectroradiometer

(MODIS) aboard NASA's Aqua satellite, including the Level-1B radiance product MYD021KM and the Level-2 cloud product MYD06 (Platnick et al., 2017), both with a spatial resolution of 1 km. The thermal infrared radiance data from MYD021KM were used for model training and testing, while the cloud properties from MYD06 were utilized for quality control, low clouds filtering and statistical analyses, as explained below.

First, we selected daytime MODIS images over the Southeast Pacific (SEP) from the first half of 2014 to create our

training dataset. After breaking them into $128 \times 128$ pixel scenes, we filtered out the cloudless scenes (cloud fraction less than 1%) and scenes containing a large amount of high clouds or ice clouds (not exceeding 10%). High clouds are defined as those with cloud top height above 6km, ice clouds are those with cloud top temperature below 273.15K. In addition, the severe stretching at the edge of MODIS granules has been avoided by filtering scenes with sensor zenith angle greater than 45°. Ultimately, these eligible scenes are manually classified as solid stratus, closed MCC, open MCC, disorganized MCC,

clustered Cu and suppressed Cu. We used the cloud properties from MYD06 product, such as cloud top height (CTH), cloud liquid water path (CLWP), cloud optical thickness (COT) and cloud effective radius (CER), to help us label and checked our results with the cloud dataset from Mohrmann et al. (2021). As a consequence, we obtained a total of 38756 labeled daytime scenes, including 3548 scenes of solid stratus, 6277 of closed MCC, 3345 of open MCC, 6739 of disorganized MCC, 8947 of clustered Cu and 9900 of suppressed Cu. These scenes were then divided into training, validation, and test dataset in a

0.6:0.2:0.2 ratio.

To classify daytime and nighttime morphological types only using one model, we utilized daytime radiance data from thermal infrared channels 29 (8.7µm), 31 (10.8µm) and 32 (12.0µm) to train our model. Notably, due to the subtle temperature variations on the cloud top, our model is unable to comprehensively discern convective cellular structures within the clouds by only depending on radiance data that reflects temperature. Thus, incorporating COT can better address the model's

shortcomings in studying these cellular structures by providing more information about the thickness of clouds. Considering that there are no nighttime COT in the Level-2 cloud product MYD06, we use the COT data retrieved by Wang et al. (2022) as the fourth channel input for our model. Their all-day COT products, obtained using a thermal infrared CNN model, have shown good consistency with both MODIS daytime products and active sensors' all-day products. In summary, the training of our model was based on daytime radiance data and COT. Once the model is well-trained, it can be generalized to nighttime

classification. The datasets used in this study are outlined in Table 1.





Table 1 Summary of datasets used in the study.

| Dataset | Count | Channels | Day/Night | Size | Period |
|---|---|---|---|---|---|
| Training dataset | 23,254 | 29,31,32, COT | Daytime | 128×128 | January– June 2014 |
| Validation dataset | 7,751 | 29,31,32, COT | Daytime | 128×128 | January– June 2014 |
| Testing dataset | 7,751 | 29,31,32, COT | Daytime | 128×128 | January– June 2014 |
| Application dataset | 18 million | 29,31,32, COT | Daytime and Nighttime | 1°×1° (128×128) | 2018–2022 |

Before starting training, we first converted the radiance data into Brightness Temperature (BT) according to the inverse Planck Function shown in Eq. (1):

$$BT(\lambda, L) = \frac{C_2}{\lambda \ln(C_1/\lambda^5 L + 1)}, (1)$$

where $\lambda$ is the wavelength (μm), L represents the radiance (W/m²·sr·μm), and $C_1 = 1.191042\times10^8$(W/m²·sr·μm⁻⁴), $C_2$ =1.4387752×10⁴ (K·μm). Then we combined BT data from the three thermal infrared channels according to the Day and Night colour scheme proposed by Lensky and Rosenfeld (2008) (Table 2). Different from the original Day and Night scheme, we did not clip each scene's data to a fixed range of maximum and minimum values because clipping might lead to the loss of important information, such as convective cell characteristics, thereby affecting model performance. Although this behavior

may lead to the data from each scene being compressed into different ranges and cause slight variations in the color of each scene image (Fig. 4), it has little impact on the model's judgment capabilities since the convolutional neural network primarily focuses on the statistical relationships between adjacent pixels in satellite images (Goodfellow, 2016). Moreover, after multiple rounds of practical training adjustments, we decided to use a factor of 2 to stretch the green channel to achieve better model prediction outcomes. The original Day and Night colour scheme and the modified scheme ultimately used in this study are

both shown in Table 2. In the end, to enhance the training efficiency and accuracy, the combined BT data and COT are normalized using Min-Max normalization following Eq. (2):

$$x' = \frac{x - min(x)}{max(x) - min(x)}, (2)$$

where $x$ represents the input data, $x'$ represents the data after normalization, $min(x)$ and $max(x)$ represent the minimum and maximum values of the input data respectively.


Earth System
Science
Data

**Table 2 The original (adapted from Lensky and Rosenfeld (2008)) and modified Day and Night color schemes**

| Color scheme | Red | | | | Green | | | | Blue | | | |
|---|---|---|---|---|---|---|---|---|---|---|---|---|
| | Channel | Min | Max | Stretch | Channel | Min | Max | Stretch | Channel | Min | Max | Stretch |
| Original Day and Night | IR12.0–IR10.8 | −4 K | 2 K | Linear | IR10.8–IR8.7 | 0 K | 6 K | $\Gamma = 1.2$ | IR10.8 | 248K | 303K | Linear |
| Modified Day and Night | IR12.0–IR10.8 | min | max | Linear | IR10.8–IR8.7 | min | max | $\Gamma = 2$ | IR10.8 | min | max | Linear |

To better match conventional climate datasets, we produced standard 1° gridded datasets by applying the trained model to a standard 1° scene, where one-degree-grid-sized satellite images were interpolated to $128 \times 128$ pixel. We further conduct some statistical analysis of meteorological conditions that may affect low cloud morphology using the co-located ERA5 reanalysis data ($1° \times 1°$, 1-hourly) from the European Centre for Medium-Range Weather Forecasts (ECMWF). Several variables, such as sea surface temperature (SST), relative humidity (RH), vertical velocity (ω) and divergence, can be directly

obtained, while lower tropospheric stability (LTS) needs to be calculated using the following equation (3):

$$LTS = \theta_{700hPa} - \theta_{1000hPa} \, , (3)$$

where θ is the potential temperature.

Furthermore, we also retrieved 5-year daytime and night-time CER ($r_e$) and COT (τ) using the CNN model from Wang et al. (2022) for subsequent statistical analysis of cloud properties. This approach will ensure the consistency of data range by

using the same cloud detection algorithm. Based on them, we can calculate the liquid water path (LWP) utilizing Eq. (4):

$$LWP = \frac{2}{3}\rho_w \tau r_e \, , (4)$$

with $\rho_w$ the density of liquid water.

**2.3 Method**

In this study, an ML model ResNet-50 (Koonce, 2021) was chosen as our model architecture. It is a deep CNN model which

employs a residual learning framework to construct a network with 50 convolutional layers. Despite a fairly deep convolutional layer, the incorporation of residual units in ResNet-50 enables direct signal transmission from earlier to later layers, ensuring high computational efficiency in deep architectures and markedly boosting both accuracy and the speed of convergence. We made some adjustments to the overall architecture of ResNet50 to better suit our datasets and the fine-tuned model structure is presented in Fig. 2a. The number of input channels was set to 4 to include the additional COT channel. Then, we configured

the output dimension of the final fully connected layer to 6 to produce a probability distribution over the 6 output classes for each scene via a softmax activation function. The internal structure of ResNet50 remains unchanged, consisting of a


preprocessing layer, four stages, and a global average pooling. The preprocessing layer includes a convolutional layer, a batch normalization (BN) layer, a ReLU activation function, and a Max Pooling layer. Each stage contains several residual blocks and is connected by skip connections (Fig. 2b).


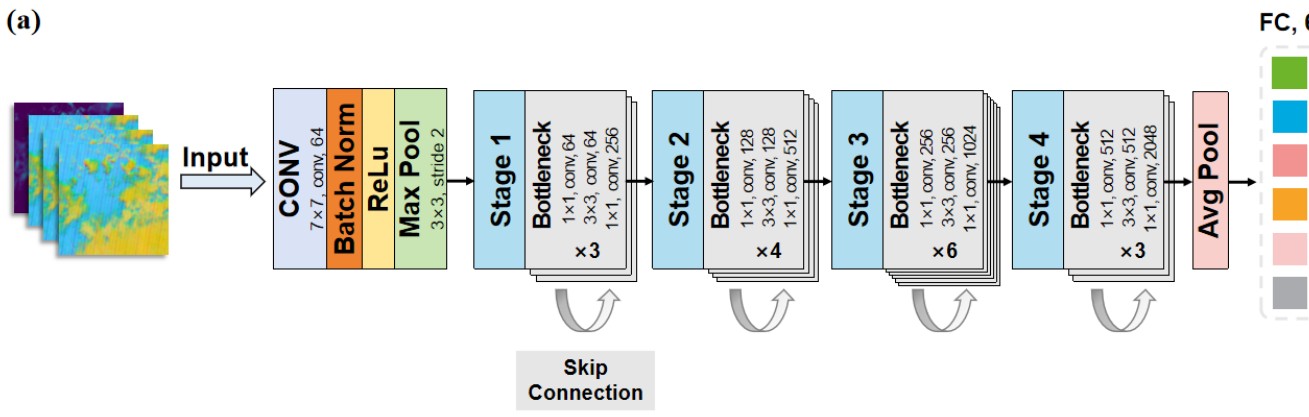

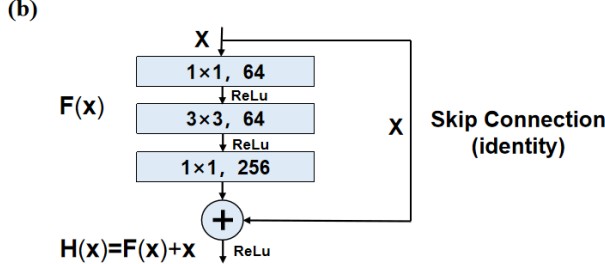

**Figure 2: (a) The fine-tuned ResNet50 model architecture. (b) The skip connection structure of the residual blocks in the model.**

Throughout the training process, we employed the Adaptive Moment Estimation (Adam) optimizer for gradient descent and utilized cross-entropy as the loss function. Given the substantial size of our training dataset, we chose a batch size of 256 to enhance memory utilization and expedite the training process. Additionally, to counteract the tendency for overfitting due to the increased noise in the radiance data of thermal infrared channels, we applied random rotation augmentation to the training images with a 50% probability. L2 regularization was also introduced to further decay the weights and prevent overfitting.





## 3 Results

### 3.1 Model performance

In this section, we present the performance evaluation of our model, as well as a classification example at night.

Our model begins to show signs of convergence around the 50th epoch, with its maximum validation accuracy reaching around 92% by the 65th epoch (Fig. S1). Compared to training without the incorporation of COT, the model's accuracy improved by 4%. Although it is a bit less accurate compared to the visible light model from Yuan et al. (2020) due to the limitations of the brightness temperature data in describing cloud morphology, it is undeniable that this model has already reached a relatively high accuracy level and it can effectively accomplish the classification tasks we proposed. The optimal model is subsequently evaluated on an independent test dataset, yielding the confusion matrix illustrated in Fig. 3. Elements on the diagonal represent the model's prediction accuracy for each type. The average precision across all types is approximately 91%. Compared with the infrared model from Lang et al. (2022), our model yields the same average prediction accuracy for closed and open MCC clouds, with closed MCC accuracy being 5% lower but open MCC accuracy being 5% higher than theirs.

In our model, closed MCC is more likely to be misidentified as solid stratus or disorganized MCC, while disorganized MCC tends to be misclassified as clustered Cu (Fig. 3). As observed in some classification samples, these misclassifications are partly attributed to the existence of mixed and transitional scenes (Fig. S2). In addition, considering the similarity between the morphology of these clouds, the misclassifications may be related to the model's limited capacity to distinguish between stratiform structures and convective cells due to the small temperature difference on the cloud top. This can also be reflected in the sample images (Fig. S3).

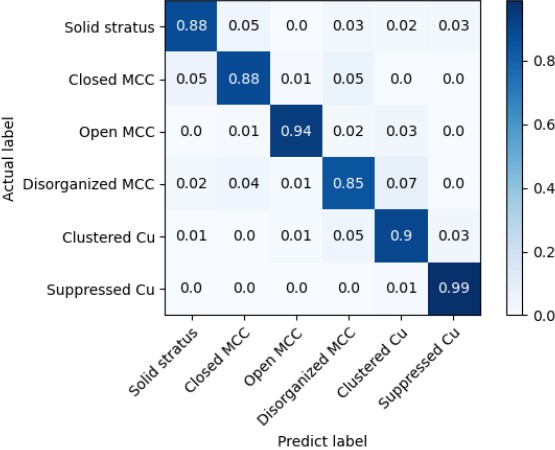

**Figure 3: The confusion matrix of the model's predictions on the test dataset. The rows of confusion matrix represent actual categories, columns represent predicted categories. The elements on the diagonal indicate the proportion of samples correctly classified by the model in each category.**





By training on daytime infrared data, this model can be applied to nighttime scenarios. Here is a successful example

shown in Fig. 4. Circles with different colors represent different cloud categories within the 1° grid. Grids without circles
indicate that they do not meet the criteria outlined in Section 2. The pseudo-RGB images are composed of thermal infrared
channels 29, 31, 32 following the modified Day and Night color scheme (Table 2). This scheme provides a clearer visual
distinction between different cloud types. However, as the data range for each image was not fixed, the colors of different
cloud types will vary in different situations. For example, in the left granule image, light yellow represents low water clouds,

green indicates thin cirrus, and dark yellow signifies thick cumulonimbus clouds. In the sample scenes of open MCC and
suppressed Cu, the yellow of low clouds becomes lighter and the green indicates small cumulus cloud. As for the remaining
four cloud types, the surrounding thin cirrus appears in green while the stratiform clouds and shallow cumulus convections are
both depicted in a brighter yellow due to their similar temperatures. Thus, it is challenging to discern convective cells among
the yellow background of stratiform clouds. That's why we incorporated COT to assist in model predictions.


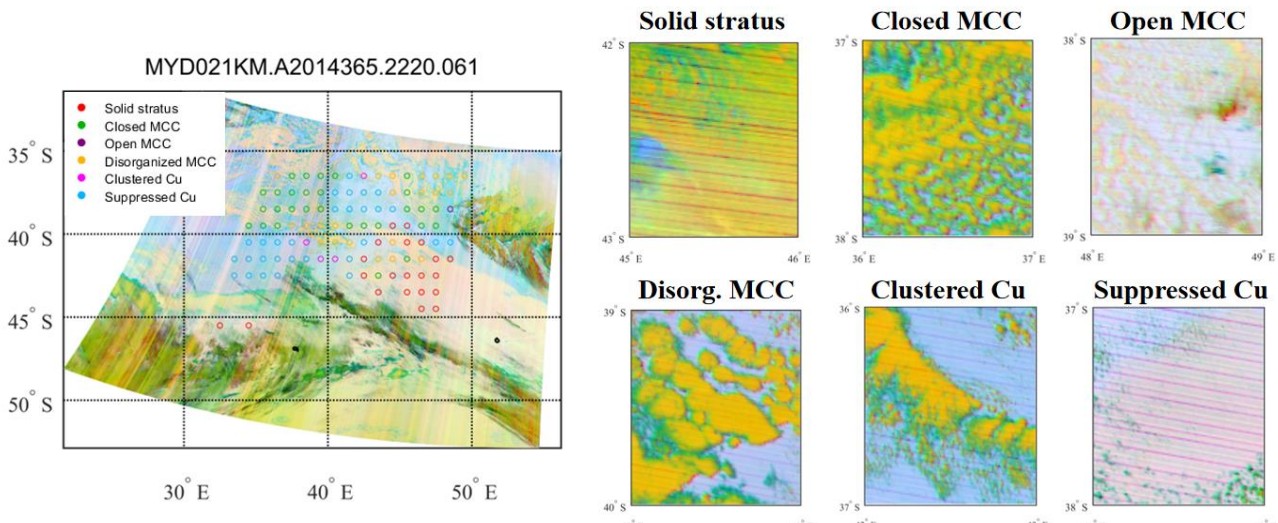

**Figure 4: A nighttime classification example for MODIS image taken at 22:20 on December 31, 2014. The pseudo-RGB images were generated from the combination of 29, 31, 32 thermal infrared channels while the classification results are derived by incorporating the retrieved Cloud Optical Thickness (COT) data.**


### 3.2 Climatology of morphological types

Using the well-trained ResNet-50 model, we classified nearly 18 million 1° MODIS scenes and recorded the occurrence counts
of different cloud types. The occurrence counts of each cloud type were divided by the total occurrences of the six cloud types
within each grid to calculate their relative frequency of occurrence (RFO). The daytime climatology of RFO for the six cloud

types is presented in Fig. 5. Each subplot's upper-right corner displays n and a percentage, n denotes the total number of

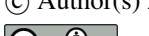


occurrences for each cloud type during the daytime over the five-year period, while the percentage indicates the proportion of each cloud type's five-year occurrences (n) relative to the five-year total occurrences of all six cloud types, which called total relative frequency.

Solid stratus predominantly distributes in nearly symmetrical latitude bands between 40°N–60°N and 50°S–60°S with a total relative frequency of 14%. In the mid to high latitudes of the Southern Hemisphere, its RFO exceeds 90%, which is higher than that in the Northern Hemisphere. Additionally, a substantial presence of solid stratus is also observed along the western coasts of continents in tropical and subtropical regions. Closed MCCs mainly appear in the cold eastern subtropical and mid-latitude oceans, with marked peak along the western coasts of North America, South America and Africa. Their total relative frequency during the daytime is relatively low, accounting for only 5%. Disorganized MCCs exhibit a distribution pattern

similar to closed MCC but are typically located farther offshore. They cover more extensive area and occur more frequently. The total relative frequency of disorganized MCC during the day is 15%, three times higher than closed MCC. In addition, it is worth noting that the peak areas of disorganized MCC are found west of closed MCC. This may be related to the transition between these two cloud types. The occurrence of open MCCs is least frequent over the global ocean, accounting for only 3%. In the waters west of Peru, there is a minor frequency peak of open MCC, which may be attribute to the fragmentation of

closed MCC caused by strong winds and precipitation (Rosenfeld et al., 2006; Eastman et al., 2022). Clustered Cu and suppressed Cu are primarily observed in tropical and subtropical regions. They have the highest overall relative frequencies, both around 30%. However, in terms of their spatial distribution, clustered cumulus is more prevalent over central and western oceans, while suppressed Cu commonly peaks in coastal waters near continents. We compared the daytime climatology with the results from Yuan et al. (2020) using visible light model and found consistent outcomes (Fig. S4).




## Climatology of daytime relative frequencies of occurrence (RFO)

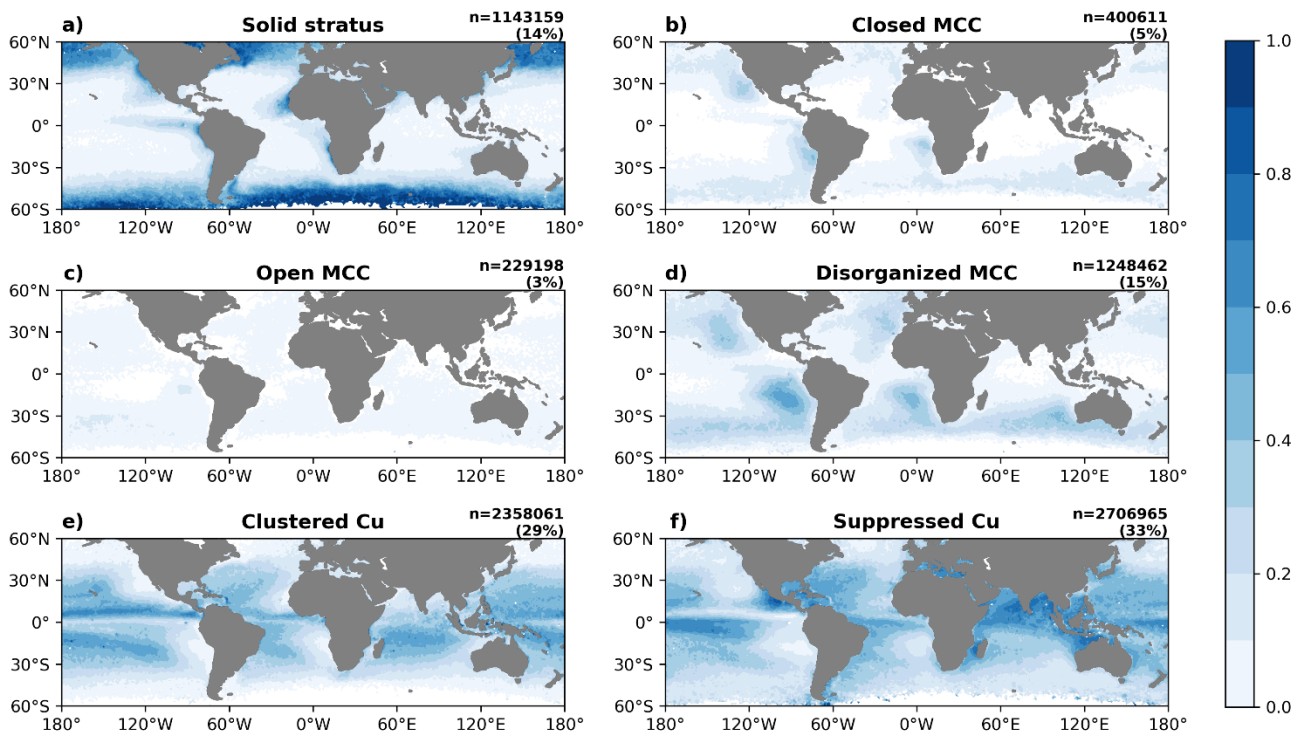

**Figure 5: The climatology of daytime relative frequencies of occurrence (RFO) for six categories from 2018 to 2022. N represents the total number of occurrences for each cloud type during the day over the five-year period, while the percentage indicates the proportion of each cloud type's five-year occurrences relative to the five-year total occurrences of all six cloud types.**


Although the spatial distribution of these six cloud types remains largely unchanged at night, their RFO show notable variations in contrast to daytime (Fig. 6). Figure 7 presents the nighttime–daytime contrast in RFO for each morphological type (nighttime minus daytime). The total nocturnal frequency of solid stratus clouds is 15%, which is similar to daytime. At night, they occur more frequently over mid-latitude oceans and less frequently in low-latitude regions (Fig. 7). The RFO for

closed MCC shows a pronounced increase at night, reaching approximately twice the levels of the day (Fig. 6). Some of the increase occurs over mid-latitude oceans, while the most significant rise is observed over the eastern subtropical ocean, particularly in the Southern Hemisphere (Fig. 7). The overall frequency of open MCC remains relatively unchanged at night, while the total frequency of disorganized MCC and clustered Cu slightly increase (Fig. 6). At night, the RFO of all these three cloud types decrease over the colder eastern subtropical and mid-latitude oceans, while increase over the warmer sea surface

at lower latitudes (Fig. 7). Notably, westward from the continents, the night-time frequency pattern of disorganized MCC exhibits an initial decrease followed by an increase. This opposite trend is most pronounced along the western coast of South

America. Among six cloud types, only the total frequency of suppressed Cu experiences a marked decline at night, with a total decrease of 11%. Explorations of the critical cloud-controlling factors contributing to these diurnal variations will be left for future work and we will only conduct a simple statistical analysis of several meteorological conditions in Section 3.5.


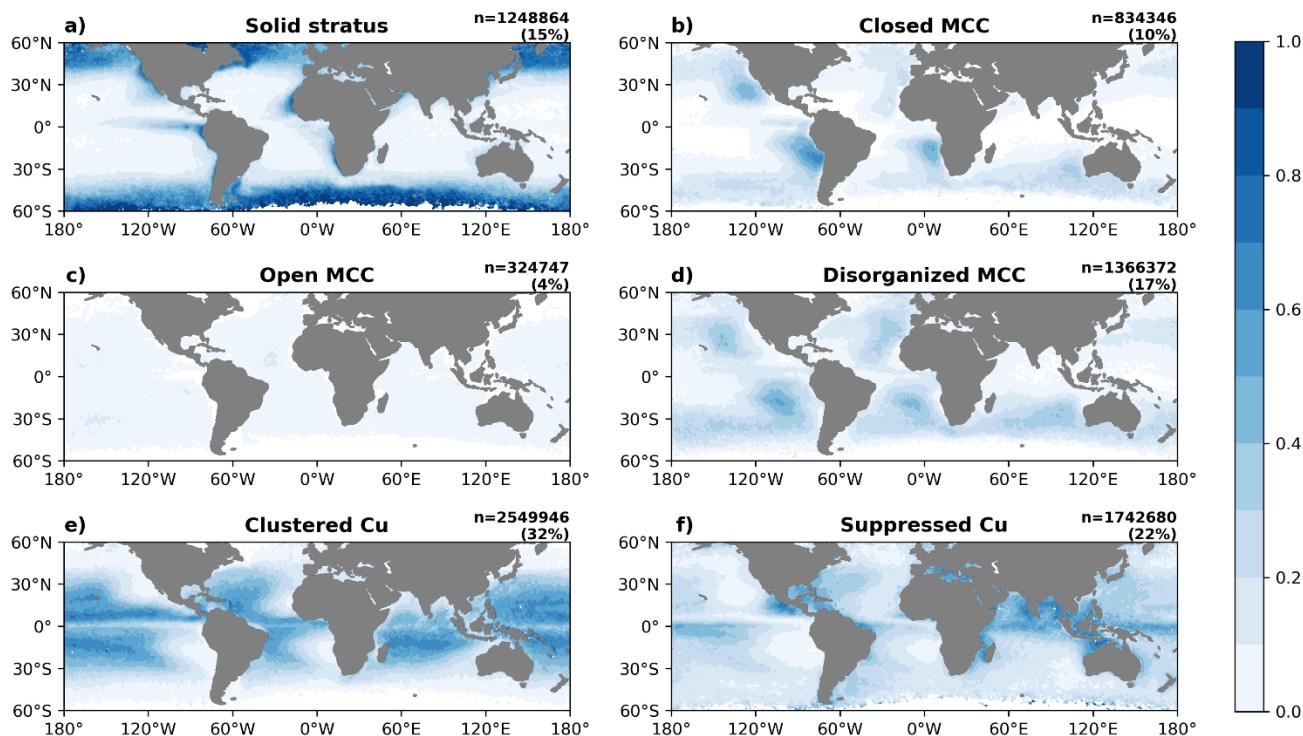

Figure 6: The climatology of nighttime relative frequencies of occurrence (RFO) for six categories from 2018 to 2022.



# Nighttime–daytime contrast of RFO climatology

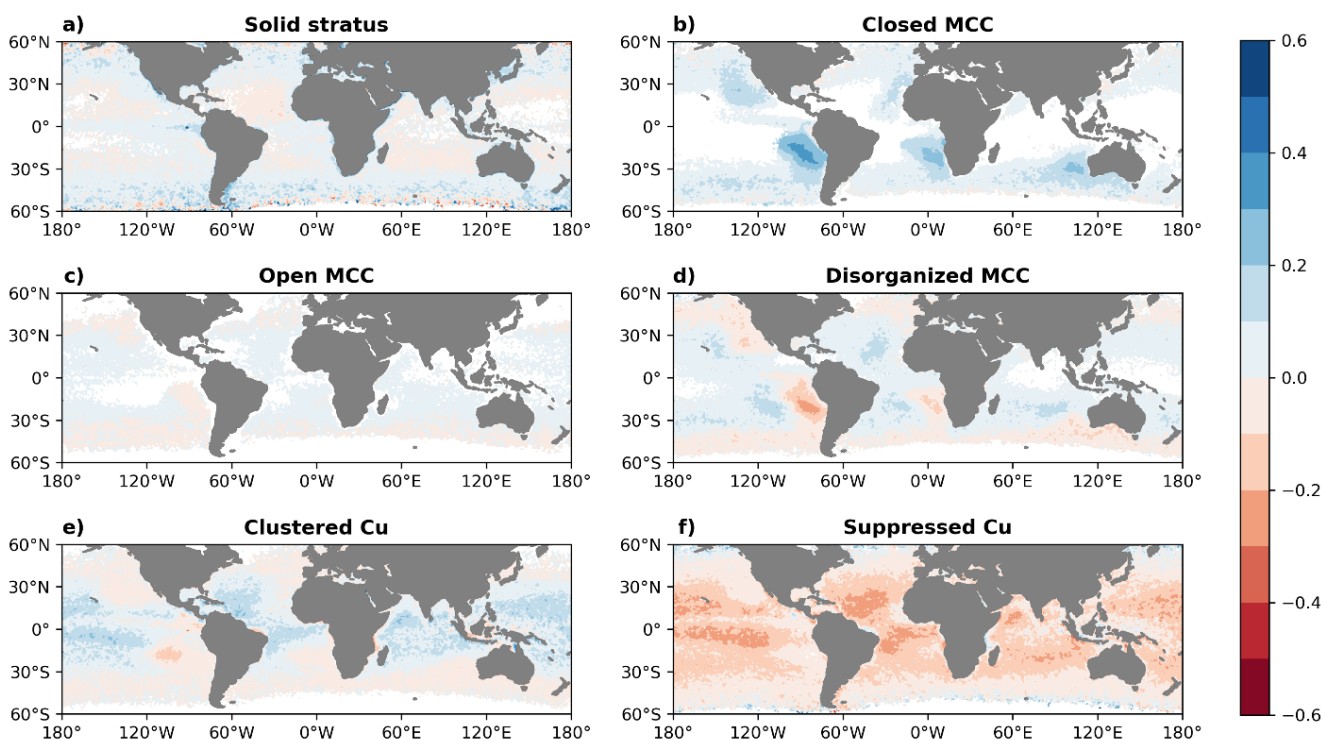

**Figure 7: The difference between daytime and night-time RFO for each morphological type (nighttime minus daytime).**

## 3.3 Seasonal variations in morphological types

We further classified the RFO of different cloud morphology by season. Figure 8 presents the seasonal variation of daytime RFO while Fig. 9 shows the nighttime situation. It can be seen from the two figures that the RFO of these six cloud types exhibit similar seasonality during both day and night. At mid latitudes, solid stratus clouds usually peak during the summer of the respective hemisphere (JJA for the Northern Hemisphere, DJF for the Southern Hemisphere) and have the lowest occurrence during the winter (DJF for the Northern Hemisphere, JJA for the Southern Hemisphere). They show equal RFO during spring and autumn in both hemispheres (MAM and SON). The peak occurrence of solid stratus in mid-latitude regions aligns with the latitudinal shift of solar insolation. Thus it can be inferred that the increased temperature and enhanced moisture availability from melting sea ice may contribute to its seasonal variation (Herman and Goody, 1976). The RFO of closed MCC notably increases during the winter (JJA) and Spring (SON) in Southern Hemisphere, particularly in the southeast Pacific (SEP) and southeast Atlantic (SEA) regions. McCoy et al. (2017) suggest that the seasonal cycle of closed MCC in such regions




correlates well with estimated inversion strength (EIS). In contrast to solid stratus, open MCC demonstrates an opposite seasonal cycle in mid-latitudes, with the highest frequency occurring in the winter of respective hemisphere (DJF for the

Northern Hemisphere and JJA for the Southern Hemisphere). Previous work suggests that its seasonality is more likely associated with cold air outbreaks in mid-latitude oceanic regions (McCoy et al., 2017). This may also explain why open MCC exhibits zonal frequency peak over the Southern Pacific during the winter of Southern Hemisphere (JJA) (Fig. 8 and Fig. 9). Disorganized MCC clouds occur more frequently over the warmer ocean surface western of the continents during the summer of respective hemisphere (JJA in the Northern Hemisphere and DJF in the Southern Hemisphere) and occur less frequently

during the winter of respective hemisphere (DJF in the Northern Hemisphere and JJA in the Southern Hemisphere). Thus, the sea surface temperature may be one of the controlling factors of its seasonal variation. All the MCC types show distinct seasonal cycles while the clustered Cu and suppressed Cu do not show marked seasonal variations during both day and night.

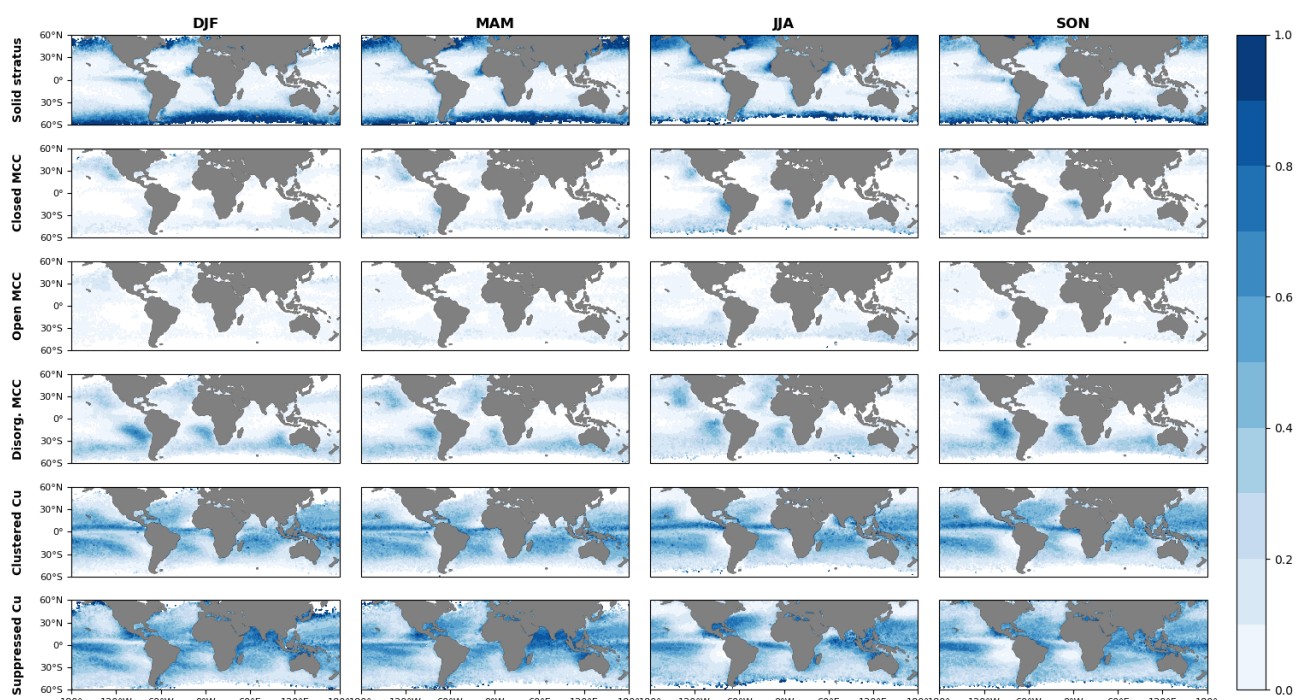

**Figure 8: Seasonal variations in daytime relative frequencies of occurrence (RFO).**

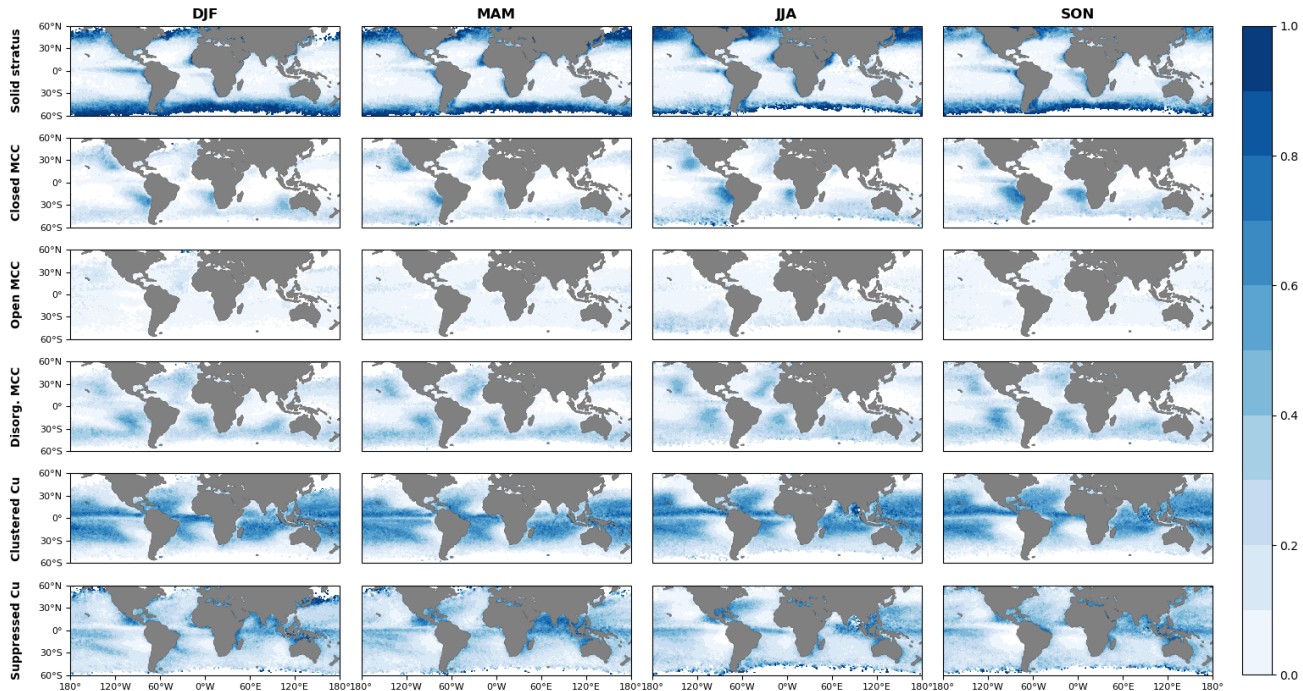

**Figure 9: Seasonal variations in nighttime relative frequencies of occurrence (RFO).**

## 3.4 Cloud properties

Different clouds types exhibit different radiative effects due to their unique physical characteristics. In Fig. 10, we compared the physical properties of each cloud type during both day and night, including cloud fraction (CF), cloud effective radius (CER), cloud liquid water path (LWP), and cloud optical thickness (COT). The CF is derived from the cloud mask in the Level-2 cloud product MYD06, while CER and COT are both retrieved using the method from Wang et al. (2022). LWP is calculated from CER and COT as mentioned in Section 2.2.

Solid stratus and closed MCC possess the highest CF, therefore the increase in their nocturnal frequency may account for a major portion of the overall rise in cloud cover. Open MCC possesses the largest CER and it will decrease by 2 µm at night. During the day, closed MCC clouds exhibit the highest values for LWP and COT. At night, their CER, LWP and COT increase further, with a substantial magnitude. The four cloud properties of disorganized MCC also show a slight increase at night.



**Figure 10: A violin plot of the properties of six cloud types over the global oceans from 2018 to 2022. (a) cloud fraction, (b) retrieved cloud effective radius, (c) cloud liquid water path, (d) retrieved cloud optical thickness. Blue represents the daytime data, and yellow represents the nighttime data. The central long dashed line in each plot represents the median of the distribution, and**
**the short dashed line indicates the interquartile range. The shape of the violin plots suggests the density distribution of the values, with wider sections indicating a higher frequency of data points.**

### 3.5 Large-scale meteorological condition

The statistics of several meteorological factors which may control the marine low cloud morphology in the Southeast Pacific
(SEP) region (0 – 30°S, 80°W – 120°W) are shown in Figure 11. The lower tropospheric stability (LTS) for the six cloud types is shown in Figure 11a. A higher LTS indicates a more stable lower troposphere. Closed MCC have the highest LTS, implying the significance of tropospheric stability in their formation. The two cumulus types have the lowest LTS because an unstable troposphere is conducive to cumulus activity. The LTS of six cloud types show different degrees of decline at night, which

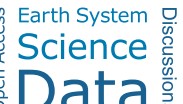

may due to the westward shift in their geographical locations. Sea surface temperature (SST) is lowest for closed MCC during

both day and night (Fig. 11b). Open MCC and disorganized MCC exhibit higher SST compared with closed MCC, which corresponds to their geographical positions. Two cumulus types have the highest SST. At night, the increase in SST of different cloud types may also be attributed to their westward movement. Figure 11c and 11d show the relative humidity (RH) at 700hPa and 1000hPa respectively. Throughout the day and night, solid stratus clouds exhibit the highest RH at both 700hPa and 1000hPa. At the 700hPa level (Fig. 11c), the RH values for two cumulus types are higher than those for MCC clouds, while at

the 1000hPa level (Fig. 11d), the difference is minimal. At night and at 700 hPa, the RH of solid stratus and two cumulus types increases, while that of closed MCC decreases. Due to lower temperatures at night, the relative humidity over sea surface for all six cloud types increases by a similar magnitude. Figure 11e indicates that all cloud types are associated with large-scale subsidence. Open MCC experience the strongest upper-level subsidence, while solid stratus has the weakest vertical motion. At night, the subsidence for all six cloud types weakens and closed MCC exhibits a more pronounced reduction. Figure 11f

presents the boundary layer anomaly divergence which calculated by subtracting the divergence at 700hPa from the surface divergence. This index has been proven effective in distinguishing between the two cumulus types (Mohrmann et al., 2021). Suppressed Cu shows the largest boundary layer anomaly divergence, indicating that strong surface divergence favors the maintenance of suppressed Cu. Clustered Cu has the smallest anomaly divergence, with weaker surface divergence. Therefore, the weakening of surface divergence at night may be the reason for the reduction of suppressed Cu.




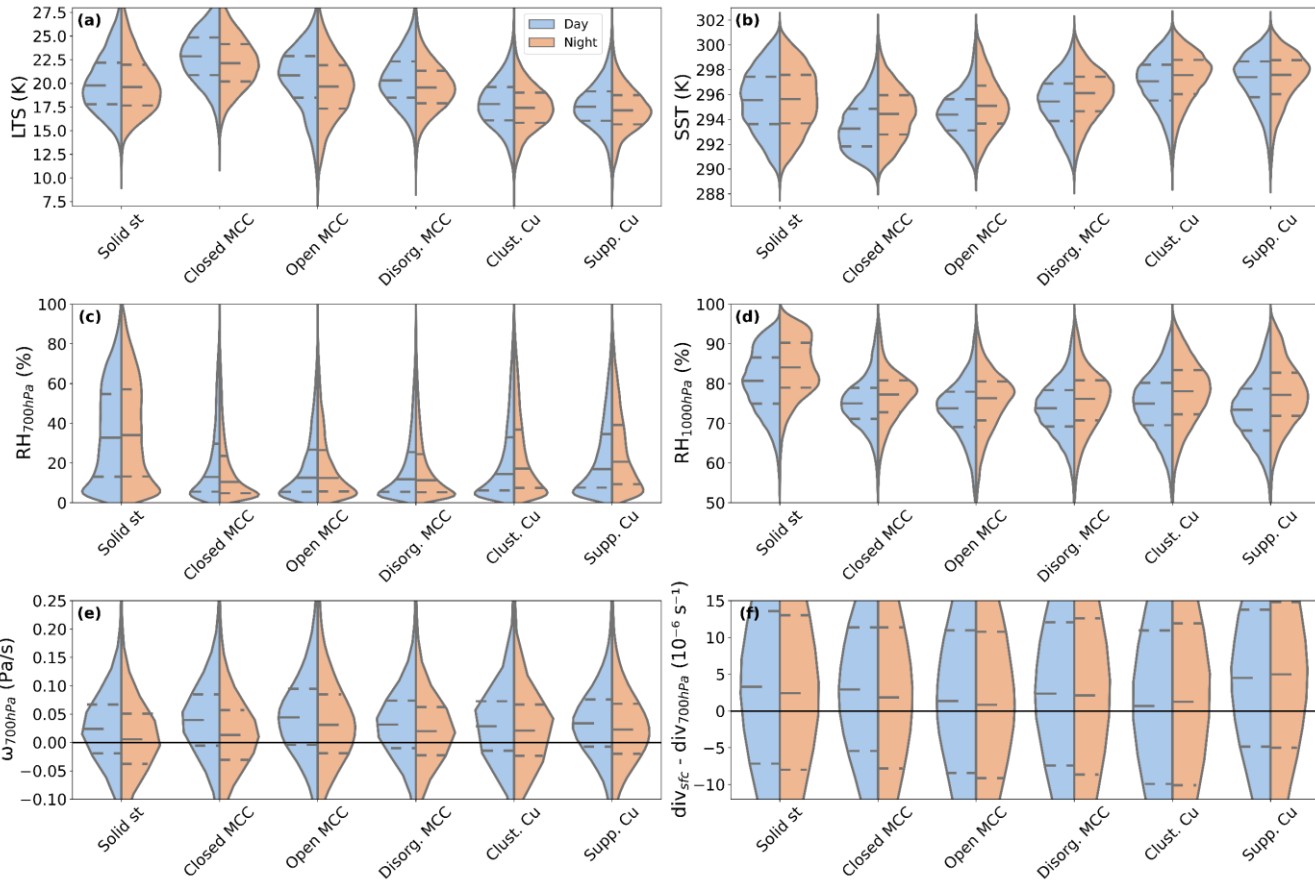

**Figure 11: Same as Fig. 10, but shows day-night comparison of meteorological conditions in the Southeast Pacific (SEP) region (0–30°S, 80°W–120°W)  from 2018 to 2022. All matched from ERA5 reanalysis data. (a) Lower tropospheric stability, (b) sea surface temperature, (c) 700hPa relative humidity, (d) 1000hPa relative humidity, (e) 700hPa vertical velocity, (f) boundary layer anomaly divergence.**

## 4 Discussion

The mesoscale cloud morphology dataset presented in this paper enables a comparative investigation of cloud morphology during both daytime and nighttime. Its 1° × 1° resolution allows for better alignment with other gridded datasets, facilitating further studies on driving factors, precipitation efficiency, and radiative effects (shortwave and longwave).

Although our model has achieved a high prediction accuracy and performed well in the classification tasks,there is still room for improvement. In future research, model performance can be optimized through these two methods: replacing the classification model and improving the quality of our training dataset. For the former one, novel deep CNN models can be applied to cloud morphology classification through transfer learning. For example, the Xception model, which achieved an accuracy of 97.66% in classifying traditional cloud types (Guzel et al., 2024), could be considered. For the latter goal, removing the mixed and mislabeled scenes from the training dataset, along with adding more representative scenes will improve the

model performance in identifying these cloud morphological types. Limitations of brightness temperature in capturing cloud-top morphology significantly impact the model's accuracy, which is a key reason why our nighttime model's performance is worse than that of Yuan et al. (2020) daytime model. However, a 4% increase in model accuracy with the inclusion of COT suggests that incorporating additional cloud property channels is another feasible approach for further enhancing our model
performance.

As we apply the model trained on $128 \times 128$ pixel scenes to the interpolated $1° \times 1°$ cloud scenes, the issue regarding to scene area require further discussion. For a given latitude, when the satellite zenith angle changes, the area of $128 \times 128$ pixel will vary due to the pixel stretching, while the area of the $1°$ grid remains constant. This is an advantage of the $1° \times 1°$ grid dataset since the larger the area is, the more possibility to cover multiple cloud types in one scene. However, at the same
satellite zenith angle, the size of $1°$ grid will change with latitude, whereas the $128 \times 128$ pixel scene area remains undistorted, which is a limitation of the $1° \times 1°$ grid dataset. Moreover, in some cases, especially with stratocumulus and cumulus clouds, interpolating images into a $1°$ grid may smooth or blur small-scale cloud features and introduce unrealistic structures that do not exist in the original images, which could lead to potential misclassifications of the model. Therefore, testing the model on a labeled, standard-grid dataset will be necessary in future work.
The six cloud types examined in this study are the most common and representative types over the ocean; however, they are not exhaustive. In future work, we will explore the overall low-cloud morphological types over the global land and ocean, and gradually extend to mid and high-level clouds.

## 5 Data Availability

Daytime and nighttime cloud classification datasets are accessible on the https://doi.org/10.5281/zenodo.13990646 (Wu et al.,
2024). The model and the code related to this article are available at https://github.com/YuanyuanWu-NJU/Cloud-morphology-dataset. MODIS data can be downloaded from NASA Official Website (https://ladsweb.modaps.eosdis.nasa.gov/). ERA5 reanalysis data are provided by ECMWF (https://cds.climate.copernicus.eu/datasets). The cloud property retrieval model of Wang et al. (2022) can be found at (https://github.com/WgQuan/cloud-property-retrievals).

## 6 Conclusion

In this study, approximately 40,000 MODIS daytime low-cloud scenes were manually labeled to train a deep residual network model, ResNet50. By using this model, we developed a new global standard-grid classification dataset (2018–2022) of marine low-cloud mesoscale morphology, encompassing classifications for both daytime and nighttime. Compared to the $128 \times 128$ pixel dataset of Yuan et al. (2020) and Mohrmann et al. (2021), our standard-grid dataset offers more uniform and widely applicable cloud morphology data, and more importantly, extending the dataset to nightly. This dataset can integrate more

easily with other climate and surface datasets, thus will provide a solid data foundation for future research on understanding cloud dynamics and their impact on climate.

The climatology of cloud morphologies is also documented. The results reveal that solid stratus dominates within the 50°–60° latitude bands in mid-latitude regions, closed MCC is most commonly found in the cold eastern subtropical and mid-latitude oceans. Disorganized MCC occurs on the warmer ocean surfaces west of closed MCC, with a much higher frequency.

Open MCC is more evenly distributed across the global oceans but with the lowest frequency. In regions with higher sea surface temperatures, such as the tropics and the trade wind zones, clustered Cu and suppressed Cu are the primary types of marine low clouds, with clustered Cu more prevalent over oceans and suppressed Cu concentrated along continental coasts. When comparing the daytime and nighttime climatology, we found that there is a pronounced increase in the RFO of closed MCC during night, whereas the occurrence of suppressed Cu undergoes a significant decline. The frequencies of disorganized

MCC and clustered Cu exhibit minor variation between day and night. From the perspective of different seasons, solid stratus and all MCC types exhibit clear seasonal cycles while two cumulus types do not show notable seasonality.

Although we did a rough statistical analysis of the meteorological factors that may affect low cloud morphology, identifying the specific dominant factors for each cloud type remains challenging, and it is left for future work that could combine the dataset we proposed. Furthermore, in the context of global warming, the long-term trends of these cloud types

during the day and night may also exhibit significant differences. The changes in Earth's radiation budget caused by them are an important component of the low cloud feedback, thus we intend to release a 20-year product for such research in the near future.

**Acknowledgments**

This work is supported by the National Nature Science Foundation of China (42475089, 41925023, 42075093, U2342223, and

430  91744208).

**Author contributions**

Y.Z. designed the study, Y.W., J.L., Y.C., Y.Z. and D.R. wrote and revised the manuscript. J.L., Y.L., Y.C. and Q.W. collected the data. J.L. and Y.Z. contributed to the modeling and data process. Q.W. and C.Z. provided the cloud property retrieval model and Y.W. retrieved the cloud property product. K.H., B.Z., Y.W., Y.L. and C.Z. contributed to interpreting results and

discussions. The entire study was conducted under the supervision of Y.Z., M.W., J.S. and D.R.

**Competing interests**

The authors declare no competing interests.





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
