# Peer review of "A Global Classification Dataset of Daytime and Nighttime Marine Low-cloud Mesoscale Morphology Based on Deep Learning Methods"

_Earth System Science Data, 2024_

## Referee Comment (RC1)

Wu et al. present a global daytime and nighttime low cloud morphology dataset classified based on deep learning methods. The work builds on the algorithms of Yuan et al. (2020; for deep learning) and Wang et al. (2022; for nighttime COT retrieval) to expand the range of cloud classification, for the first time, to nighttime retrievals. The dataset is novel, unique, and of high quality, with significant potential for use in cloud and climate studies. However, I find that the manuscript lacks some important information, particularly regarding their choice of testing data, sensitivity studies, and data screening techniques. Moreover, the data could be presented in a better way for a wider scientific community. Therefore, I recommend a major revision of the manuscript/dataset by addressing the following comments before it can be considered for potential acceptance in *ESSD*.

**Comments:**

1. While the TIR-CNN-based retrieval of cloud properties in Wang et al. (2022) could be a better one compared to the TIR-based algorithm, it cannot replace the standard daytime retrieval algorithm in MODIS. Therefore, to justify using the TIR-CNN-based COT used in training the MCC-classification, I suggest the authors include an additional validation in this study, which includes comparing their MCC classification with that from the outputs of a CNN trained on MODIS daytime COT. Even though I agree that the choice of TIR-CNN-based COT is methodologically justified to stay consistent in their application to both daytime and nighttime retrievals, the inclusion of this additional validation/sensitivity study will strengthen their results and the MCC classification dataset.

2. The authors use different numbers of samples in each MCC category to train the model. For example, (0.6 times) 9,900 labeled suppressed Cu are used compared to just (0.6 times) 3,548 solid stratus samples. Shouldn't this disparity impact the performance of the classification for different MCC categories? Can the authors comment on this?

3. Why do the authors interpolate the data within a 1°X1° scene to 128 X 128 pixels? Even if I consider the finest resolution of 1km, the number of pixels within a 1°X1° scene would be less than 128X128 pixels, leading to extrapolation-related truncation error. Also, did the authors perform any sensitivity test regarding the size of the scene considered in training the model except from 1°X1°? Wood and Hartmann (2006) use native MODIS 256 X 256 pixels in their classification. Increasing the grid size may reduce the probability of misclassifications (Fig. 3). For instance, considering a smaller domain may result in misclassification of edges of open cells into clustered Cu. In case you achieve a better classification, the resulting dataset can be resampled to a finer grid easily for future use in conjunction with other climate and weather-related datasets.

4. Information is missing regarding why channels 29, 31, and 32 were particularly used in training and classification when multiple other cloud-top-related channels (33-36) are available in MODIS.

5. What are the parallel yellow and red lines in the panels of Figure 4? Are these physical and being used in classification or graphics-related artefacts?

6. Regarding the dataset, I highly recommend using standard data formats used in atmospheric sciences like netCDF and HDF for easy cross-platform and cross-software accessibility. Not all users will be accustomed to the Python-specific NumPy format.

7. Since this is a data-descriptor paper, some important information on the contents (variables and how they are calculated) and the file nomenclature should be included in the manuscript. It may be presented as a separate sub-section within Section 2 and summarized using an additional table. This information is currently missing from the manuscript.

**Minor comments:**

1. Line 49-51: More recently Goren et al. (2019) showed a similar delay in closed-to-open transition using LES.

2. Line 60-61: The cloud morphology dataset by Wood and Hartmann (2006) has been expanded to more than a decade of MODIS observations, the Morphology Identification Data Aggregated over the Satellite-era (MIDAS), by McCoy et al. (2023).

3. Line 64: Abbreviation VGG not defined!

4. Line 64: "... for daytime scenes ...". All the morphology datasets discussed prior to this point correspond to daytime observations, don't they?

5. Line 102-103: "Disorganized MCC ... larger droplets and lower optical thickness." Can the authors cite studies that have demonstrated this fact?

6. Line 106: Citation missing!

7. Line 116: "spatial resolution of 1 km" This resolution is for nadir pixels. It changes with sensor zenith angle.

8. Line 121: The authors state that they filter out scenes with more than 10 % high clouds or ice clouds. How do the authors deal with ice/high cloud pixels in scenes where they are less than 10%? Are they set to missing values and not used in either training or classification steps?

9. Line 172: How is the reanalysis data co-located? Do you select the nearest timestamp or interpolated the data to MODIS observations?

10. A link to the classification dataset is missing in the "Data Availability" section.

11. No information on the file "example.xlsx" in the data repository.

**Language-related suggestions:**

Line 21: Abbreviation RFO defined in abstract is not used.

Line 84: dependent?

Line 91: Prior to "Section 2 intro…", perhaps insert an introductory sentence like "The manuscript is organized as follows."

Line 184: Abbreviation ML is not defined

Line 210: Remove underscore after Yuan et al. (2020)

Line 383: Consider changing the word "worse"

Line 409: "… nightly …" Do you mean nighttime?

**References**:

Goren, T., Kazil, J., Hoffmann, F., Yamaguchi, T., & Feingold, G. (2019). Anthropogenic air pollution delays marine stratocumulus break-up to open-cells. Geophysical Research Letters, 46, 14135–14144. https://doi.org/10.1029/2019GL085412

McCoy, I. L., McCoy, D. T., Wood, R., Zuidema, P., & Bender, F. A.-M. (2023). The role of mesoscale cloud morphology in the shortwave cloud feedback. *Geophysical Research Letters*, 50, e2022GL101042. https://doi.org/10.1029/2022GL101042

Wang, Q., Zhou, C., Zhuge, X., Liu, C., Weng, F., and Wang, M.: Retrieval of cloud properties from thermal infrared radiometry using convolutional neural network, Remote Sensing of Environment, 278, 113079, https://doi.org/10.1016/j.rse.2022.113079, 2022.

Yuan, T., Song, H., Wood, R., Mohrmann, J., Meyer, K., Oreopoulos, L., and Platnick, S.: Applying deep learning to NASA MODIS data to create a community record of marine low-cloud mesoscale morphology, Atmos. Meas. Tech., 13, 6989-6997, https://doi.org/10.5194/amt-13-6989-2020, 2020.

---

## Referee Comment (RC3)

Review of "A Global Classification Dataset of Daytime and Nighttime Marine Low-cloud Mesoscale Morphology Based on Deep Learning Methods" by Wu et al. [MS number: essd-2024-536]

This study produces a global dataset of daytime and nighttime low-cloud mesoscale morphologies (categorized into six types) using a convolutional neural network through a combination of MODIS infrared radiance data and machine-learning-retrieved cloud optical thickness. Leveraging this novel dataset, the authors analyzed the day-night contrast in climatology, seasonal cycles, and cloud properties of cloud morphologies. One of notable findings is the significant diurnal variation in the occurrence frequency of closed MCC and suppressed Cu. The primary contribution of this work lies in the generation of nighttime low-cloud morphology data, which complements the well-established daytime morphology datasets from prior studies. This advancement would inspire and enable more downstream research like understanding the diurnal cycle of cloud morphology and cloud-longwave-radiation-climate feedback. The manuscript is overall well-written and well-organized, with nice presentation of figures. However, my major concern pertains to the limitations in the model's training and validation processes, which could impact the dataset's reliability. Addressing these issues would significantly strengthen the study's contribution to the marine low-cloud research community. I'd like to recommend a major revision before this manuscript is considered for publication in ESSD.

**Major comments:**

1. One of my primary concerns is the validity of applying a regionally trained deep learning (DL) model to global predictions. In this study, the authors developed their model using data from the SEP region only and then applied it to generate a global dataset. While the model demonstrates relatively high prediction accuracy over SEP (Figure 3), it is unclear whether this performance extends to global applications. Regarding this issue, the authors should first clarify the rationale for selecting SEP as the training region rather than using a global or other regional dataset. Was this choice subject to the limited availability of the data, or is there a similarity in morphology climatology between SEP and the global scale? If SEP is your best choice at the moment, it would be essential to evaluate whether using a regionally trained model for global predictions is reasonable. One approach to examine this would be to generate a global map of prediction accuracy for each cloud morphology type to check the model's global performance. Additionally, the authors could examine the differences in the PDFs of thermal radiance, COT, and cloud morphology between SEP and the global dataset. A smaller difference or larger overlaps would indicate less extrapolation by the model, enhancing the credibility of the global dataset.

Similarly, the authors would have to be careful when extending the daytime-trained model to nighttime predictions, as this may also introduce potential extrapolation issues. The authors provided only a single example to illustrate the model's success at nighttime, which is insufficient to establish its statistical reliability. To address this concern, additional cases should be analyzed to validate the model's nighttime performance. Alternatively, examining the differences in the PDFs of thermal radiance, COT, and cloud morphology between daytime and nighttime could help assess the extent of extrapolation and ensure the robustness of the predictions.

2. Regarding the model training, validation, and testing, the data-splitting strategy is unclear. For instance, was the dataset split randomly or manually into the 6:2:2 ratio? Furthermore, the validation method used to assess the model's predictions has not been described. The authors should clarify these aspects to improve the robustness of their results.

3. Given the critical role of cloud morphologies in Earth's radiation budget, the authors could consider including a climatological analysis of shortwave and longwave radiation at the TOA for the six cloud morphology types. Adding such an analysis would significantly enhance the insights and scientific value of this study.

**Minor comments:**

L30: longwave warming effects are more significant for high clouds, which might not be so for low clouds.

L67: What is the major difference between the six-type classification of this study and the four-type one here?

L81: Do you mean the decline in the *long-term* trend?

L83: "how much they contribute to … remain unclear" to "how nighttime cloud cover varies under different cloud morphology types remain unclear."

L90: Please clarify the temporal and spatial resolution.

L97: "created" to "driven"

L119: Please clarify the temporal resolution of the training dataset.

L121-122: Have you excluded middle clouds (i.e., those situated between 3 and 6 km)? These clouds are prevalent over midlatitude oceans, and they also contaminate low cloud observations.

L174: Please clarify the level of the divergence used.

L199: I'd suggest labeling the input variables (three channels and COT) and the output variables (six cloud morphology types) in Figure 2a to improve its clarity and readability.

L210: It looks like the improvement is limited. Have you examined the COT retrieval uncertainty? If it is greater than the improved accuracy, it would be unnecessary to include the COT into the predictors.

L210: Typo: "Yuan et al. (2020)_due to" to "Yuan et al. (2020) due to"

L212: Which is it relative to?

L219: "clustered Cu" to "clustered Cu or closed MCC"

L250: "n denotes" to "with n denoting"

L305: "its seasonal variation" to "the peak in summer"

L332: do you mean "decrease by 2 microns *on average*"?

L333: Please clarify whether the LWP mentioned here represents the in-cloud value or the grid-box mean value.

L349: Why is there a westward shift at night? Also, for stratocumulus clouds, LTS is usually higher at night. Why does it decline for closed MCC at night?

L351: It would be more interesting to discuss their physical reason.

L367: Why are the results shown here only for SEP, while Figure 10 presents global results?

---

## Author Comment (AC1)

Dear Editor,

We are grateful for the careful reading, helpful comments, and constructive suggestions of the reviewers, which have allowed us to clarify and improve the manuscript. The three reviewers have provided valuable insights from different perspectives. Reviewer #1 focused on the quality and application of the dataset, particularly the selection and presentation of the data. Reviewer #2 emphasized the quality of the training, validation and testing datasets, as well as the reliability assessment of the nighttime results. Reviewer #3 concentrated on the model's training and validation, as well as the application of the cloud dataset.

In response to their comments, we made the following main changes and improvements in our manuscript:

1. We have conducted a sensitivity experiment comparing the TIR-CNN-based COT with the MODIS daytime COT, and the results are now included in the revised manuscript to strengthen our cloud dataset.

2. All the NumPy format files were converted to HDF format to enhance the cross-platform and cross-software accessibility of our dataset.

3. We have added a new sub-section in Section 2 to describe the dataset contents, including variables, their calculation methods, and the file nomenclature, along with an additional table to summarize.

4. We validated the representativeness of our training dataset by examining the differences in the probability distribution functions (PDFs) of thermal radiance, cloud optical thickness (COT), and cloud morphology between our dataset and the global full-year daytime dataset. Additionally, we compared the PDFs of thermal radiance, COT, and cloud morphology between daytime and nighttime to further verify the reliability of the nighttime results.

5. The detailed descriptions of the validation method and model performance metrics were added in the manuscript, including precision, F1-score, and recall, to enhance the reliability and robustness of our model.

Although some of the improvements suggested by the reviewers are not reflected in this paper, many of the directions they pointed out are being actively pursued by other members of our team. This cloud dataset serves as a foundational work, and we hope that, under its guidance, a series of studies will be conducted in this direction, encouraging the community to pay more attention to the research in this field.

Below we addressed the reviewers' comments, with the reviewer comments in black and our response in blue. The revised sentences in the manuscript are indicated in italics.

Yannian Zhu

On behalf of the co-authors

**Reviewer #1**

Wu et al. present a global daytime and nighttime low cloud morphology dataset classified based on deep learning methods. The work builds on the algorithms of Yuan et al (2020; for deep learning) and Wang et al (2022; for nighttime COT retrieval) to expand the range of cloud classification, for the first time, to nighttime retrievals. The dataset is novel, unique, and of high quality, with significant potential for use in cloud and climate studies. However, I find that the manuscript lacks some important information, particularly regarding their choice of testing data, sensitivity studies, and data screening techniques. Moreover, the data could be presented in a better way for a wider scientific community. Therefore, I recommend a major revision of the manuscript/dataset by addressing the following comments before it can be considered for potential acceptance in ESSD.

We appreciate your recognition of the novelty and quality of our dataset. Regarding to your concerns, we realize the need for further clarification and improvement. In the revised manuscript, we have provided additional details on the selection criteria for the testing data, conducted sensitivity studies to evaluate the robustness of our model, and further clarified our data processing and screening methods. Furthermore, we have improved the presentation of our dataset by providing additional descriptions and modifying its storage format, which significantly enhances its accessibility. We believe these revisions will address your concerns and improve the quality of our manuscript. Please see the response below for further details.

**Comments:**

1. While the TIR-CNN-based retrieval of cloud properties in Wang et al (2022) could be a better one compared to the TIR-based algorithm, it cannot replace the standard daytime retrieval algorithm in MODIS. Therefore, to justify using the TIR-CNN-based COT used in training the MCC-classification, I suggest the authors include an additional validation in this study, which includes comparing their MCC classification with that from the outputs of a CNN trained on MODIS daytime COT. Even though I agree that the choice of TIR-CNN-based COT is methodologically justified to stay consistent in their application to both daytime and nighttime retrievals, the inclusion of this additional validation/sensitivity study will strengthen their results and the MCC classification dataset.

Thank you for your valuable suggestion. We agree that an additional validation comparing the algorithms based on TIR-CNN-based COT and MODIS daytime COT would be beneficial. In response, we have performed a sensitivity experiment by training a CNN model using MODIS COT and the original three infrared channels. The results indicate that the accuracy of the CNN model based on MODIS COT is 91.3%, which is almost identical to the 91.5% accuracy achieved by the model using the COT retrieved from Wang et al. (2022). The new training results based on MODIS

COT are shown in Figure R1. This demonstrates that the TIR-CNN-based COT can effectively serve as a reliable alternative to MODIS COT for cloud classification, which further reinforce the reliability of our model and the MCC classification dataset. Therefore, we have added some supplementary statements regarding this sensitivity experiment in the '2.2 Data' Section, as follows: "*To validate the reliability of using TIR-CNN-based COT as a replacement for MODIS COT, we conducted a sensitivity experiment: comparing our classification with the outputs of a CNN trained on MODIS daytime COT. The results (Fig. S3) showed that the accuracy of both models is nearly identical, indicating that TIR-CNN-based COT is a reliable alternative to MODIS COT.*" (Lines 142-145).

[Figure]

**Figure R1.** Model training results based on MODIS COT. The above figure shows the model's accuracy on the training and validation datasets, while the figure below displays the confusion matrix obtained for the model on the test dataset.

2. The authors use different numbers of samples in each MCC category to train the model.

For example, (0.6 times) 9,900 labeled suppressed Cu are used compared to just (0.6 times) 3,548 solid stratus samples. Shouldn't this disparity impact the performance of the classification for different MCC categories? Can the authors comment on this?

We agree that the disparity in the number of samples across different cloud types can potentially affect the model's performance. When there is an imbalance in the training data, the model might become biased towards the categories with larger sample sizes and perform less accurately on the underrepresented categories.

However, since each category in our training set contains a sufficient number of samples and the sample size ratio between categories does not exceed 4:1, the impact of the sample imbalance on our training should be relatively small.

To approve this, we conducted an experiment in which 2,000 samples were randomly selected from each category to train a new CNN model. The training result (Figure R2) shows a prediction accuracy of 88.8%, which is slightly lower than our previous result. This may be due to the relatively small sample size. Despite this, it indicates that sample imbalance has little impact on our training process. Therefore, we have added some explanations in Section '2.2 Data': *"Despite the disparity in sample sizes within our training dataset, it has been demonstrated to yield superior model performance compared to training on a balanced dataset (Fig. S2)."* (Lines 132-133)

In the future, we plan to add more manually labeled samples to the category with the fewest samples to explore whether a larger balanced training dataset can lead to a better performance.

[Figure]

**Figure R2:** Model training results with 2,000 samples for each category. The above figure shows the model's accuracy on the training and validation datasets, while the figure below displays the

confusion matrix obtained for the model on the test dataset.

3. Why do the authors interpolate the data within a 1º × 1º scene to 128 × 128 pixels? Even if I consider the finest resolution of 1km, the number of pixels within a 1º × 1º scene would be less than 128 × 128 pixels, leading to extrapolation-related truncation error. Also, did the authors perform any sensitivity test regarding the size of the scene considered in training the model except from 1º × 1º? Wood and Hartmann (2006) use native MODIS 256 × 256 pixels in their classification. Increasing the grid size may reduce the probability of misclassifications (Fig. 3). For instance, considering a smaller domain may result in misclassification of edges of open cells into clustered Cu. In case you achieve a better classification, the resulting dataset can be resampled to a finer grid easily for future use in conjunction with other climate and weather-related datasets.

Thank you for pointing out these valuable issues. For the first question, we did not perform extrapolation; rather, we performed refinement, where the 128 × 128 pixels images obtained through interpolation are included within a 1-degree grid. We have made the following revisions in the manuscript to clarify the translation: "*To align with conventional climate datasets, we developed a standard 1° gridded datasets by applying the trained model to 1°-resolution images, where the 1°×1° satellite images were interpolated and refined to 128 × 128 pixels.*" (Lines 174-175)

Although this may still lead to issues near the cloud edges, it avoids interference from other grid pixels. Particularly at higher latitudes, a 128 × 128 pixels scene will cover multiple grids, thus the classification of center grid may be interfered by the cloud features in neighboring grids. By directly classifying the cloud pixels within a 1° grid, this type of error can be minimized.

For the secondary question regarding to the sensitivity test of the scene size, first, the size of the scene can influence the definition of cloud types. A larger scene may necessitate a redefinition of cloud types, such as the Sugar, Gravel, Fish, and Flowers categories defined by Stevens et al. (2020) (10º ×10º). Secondly, Yuan et al. (2020) also noted that larger scene sizes increase the probability of multiple different cloud types appearing within a single scene, while smaller scenes may lack sufficient contextual information for effective classification. Beyond the consideration of Yuan et al. (2020), we took additional factors into account and ultimately decided to adopt a standardized grid scene to better address the issue of pixel stretching at high latitudes. Moreover, standardized cloud classification datasets are more convenient for the community to use, as most current studies utilize 1-degree grid datasets for analysis.

Thirdly, we agree that using a larger scene size helps to constrain and reduce the misclassifications caused by smaller domains. Therefore, a more reasonable approach would be classifying the integration of scenes of different sizes using the automatic unsupervised learning, which is an area we plan to explore in our future work.

4. Information is missing regarding why channels 29, 31, and 32 were particularly used in training and classification when multiple other cloud-top-related channels (33-36) are available in MODIS.

Channels 29, 31, and 32 were chosen because they effectively represent cloud properties and cloud-top temperature, which are critical for cloud classification. Specifically, channels 29 (8.7μm) is sensitive to water-vapour absorption, while Channels 31 (10.8μm) and 32 (12.0μm) provide valuable information on cloud-top temperature. In contrast, Channels 33-36 are more focused on cloud-top altitude and other related properties. We previously conducted an experiment using all six channels (29, 31, 32, 33, 34, and 35) for model training, the results were very similar to those obtained only using channels 29, 31, 32 (unfortunately, the experimental data from that trial was not properly archived). In order to reduce the amount of data, we ultimately chose the three channels 29, 31, 32 as inputs.

Therefore, we added explanations in the manuscript to justify our selection of these three channels: *"Thermal infrared (TIR) channels 29 (8.7μm), 31 (10.8μm) and 32 (12.0μm) were specifically chosen as they most effectively represent the cloud properties and cloud-top temperature."* (Lines 135-136)

5. What are the parallel yellow and red lines in the panels of Figure 4? Are these physical and being used in classification or graphics-related artefacts?

The striped patterns (yellow and red lines) visible in Figure 4 are graphical artifacts generated by satellite sensors. Although these artifacts may create some visual noise, they have minimal effect on our pattern identification process since our model relies on qualitative pattern recognition rather than quantitative analysis. Furthermore, the CNN model can filter the noise out, so we opted not to eliminate the striped noise during the training and classification processes.

Nevertheless, we also tried several methods to eliminate this noise and improve the visual quality, such as mean filtering, Fourier transform, and directional filtering. However, the stripe noise in our data is not traditionally horizontal or vertical, and there are no significant numerical characteristics, so none of these methods were effective in removing it. When the noise was removed through these methods, the image became very blurry, as shown below. Perhaps AI-based methods could help eliminate it, and we will continue to try that.

[Figure]

**Figure R3:** The image processed with directional filtering.

6. Regarding the dataset, I highly recommend using standard data formats used in atmospheric sciences like netCDF and HDF for easy cross-platform and cross-software accessibility. Not all users will be accustomed to the Python-specific NumPy format.

We fully agree that using standard data formats would enhance cross-platform and cross-software accessibility. In response, we have converted all the files previously in NumPy format to HDF files.

7. Since this is a data-descriptor paper, some important information on the contents (variables and how they are calculated) and the file nomenclature should be included in the manuscript. It may be presented as a separate sub-section within Section 2 and summarized using an additional table. This information is currently missing from the manuscript.

Indeed, our manuscript lacks a description of the dataset-related content. We have added the following sub-section in Section 2 to explain the contents of our dataset:

*"2.3 Marine Low-cloud Mesoscale Morphology Dataset*
*Our cloud dataset provides global classifications of daytime and nighttime marine low-cloud mesoscale morphology for the years 2018-2022, with a spatial resolution of 1° × 1° and a temporal resolution of 5 minutes. The dataset is provided in two kinds of files: those prefixed with "day" store the daytime classification results for each year, while files with the prefix "night" contain the nighttime classification results for each year. Both sets of files include the same variables. Table 3 provides an overview of the variables and their associated information. The key variables in the dataset include 'date' (representing the time of the 1° × 1° scene, formatted as the MODIS granule date), 'lon' and 'lat' (indicating the central longitude and latitude), and 'cat' (assigned cloud category, the values from 0 to 5 correspond to 'Solid Stratus', 'Closed MCC', 'Open MCC', 'Disorganized MCC', 'Clustered Cu', and 'Suppressed Cu', respectively). Additionally, 'cert' represents the model certainty, quantifying the probability that the cloud morphology belongs to the assigned category. 'low_cf' denotes the low cloud fraction, and 'COT_CNN', 'CER_CNN', and 'LWP_CNN'*

*provide the in-cloud average cloud optical thickness, effective radius, and liquid water path respectively, as derived from the TIR-CNN model from Wang et al. (2022). The 'Sensor_zenith' variable indicates the scene average sensor zenith angle.''* (Lines 190-202)

*Table 3 Variables of the Daytime and Nighttime Global Marine Low-cloud Mesoscale Morphology Dataset*

| Variable Name | Description | Source | Spatial Resolution | Temporal Resolution | Units |
|---|---|---|---|---|---|
| *date* | *Time of the 1°×1° grid point, formatted as 'YYYYDDD.HHHH'* | *MODIS MYD021* | *1 ° ×1 °* | *5 minutes* | *-* |
| *lon* | *Central longitude (-180,180)* | *MODIS MYD021* | *1 ° ×1 °* | *5 minutes* | *degree (°)* |
| *lat* | *Central latitude (-60,60)* | *MODIS MYD021* | *1 ° ×1 °* | *5 minutes* | *degree (°)* |
| *cat* | *Category of the cloud morphology: 0-Solid stratus, 1-Closed MCC, 2-Open MCC, 3-Disorganized MCC, 4-Clustered Cu, 5-Suppressed Cu* | *Cloud Classification Model* | *1 ° ×1 °* | *5 minutes* | *-* |
| *cert* | *Model certainty* | *Cloud Classification Model* | *1 ° ×1 °* | *5 minutes* | *-* |
| *low_cf* | *Cloud fraction of low clouds* | *MODIS MYD06* | *1 ° ×1 °* | *5 minutes* | *-* |
| *COT_CNN* | *In-cloud average cloud optical thickness (COT)* | *TIR-CNN model of Wang et al. (2022)* | *1 ° ×1 °* | *5 minutes* | *-* |
| *CER_CNN* | *In-cloud average cloud effective radius (CER)* | *TIR-CNN model of Wang et al. (2022)* | *1 ° ×1 °* | *5 minutes* | *μm* |
| *LWP_CNN* | *In-cloud average cloud liquid water path (LWP)* | *Calculated from COT_CNN and CER_CNN* | *1 ° ×1 °* | *5 minutes* | *g/m²* |

| Sensor_zenith | Scene average sensor zenith angle | MODIS MYD021 | 1°×1° | 5 minutes | degree (°) |
|---|---|---|---|---|---|

**Minor comments:**

1. Line 49-51: More recently Goren et al (2019) showed a similar delay in closed-to-open transition using LES.

Yes, references have been added.

2. Line 60-61: The cloud morphology dataset by Wood and Hartmann (2006) has been expanded to more than a decade of MODIS observations, the Morphology Identification Data Aggregated over the Satellite-era (MIDAS), by McCoy et al (2023).

Thank you for reminding us! We have revised the sentence to: *"Their work was pioneering and has since been extended to more than a decade of MODIS observations by McCoy et al. (2023)"* (Line 62)

3. Line 64: Abbreviation VGG not defined!

The abbreviation *"VGG"* has already been defined as *"Visual Geometry Group"* in the manuscript.

4. Line 64: "… for daytime scenes …". All the morphology datasets discussed prior to this point correspond to daytime observations, don't they?

Yes, they did. We have removed the original sentence *"Their dataset has higher spatial resolution, at 128×128 pixel, but also only includes classifications for daytime scenes."*

5. Line 102-103: "Disorganized MCC … larger droplets and lower optical thickness." Can the authors cite studies that have demonstrated this fact?

The citations for this fact have been added, and the original sentence has been checked and revised to: *"Disorganized MCC are a mix of convective elements and extensive stratiform clouds, marked by smaller droplets and lower optical thickness (Yuan et al., 2020; Liu et al., 2024)."* (Lines 104-105)

6. Line 106: Citation missing!

References have been added.

7. Line 116: "spatial resolution of 1 km" This resolution is for nadir pixels. It changes with sensor zenith angle.

Thank you for pointing that! The original sentence has been further clarified as: *"The primary observation data utilized in this study were derived from the Moderate*

*Resolution Imaging Spectroradiometer (MODIS) aboard NASA's Aqua satellite, including the Level-1B radiance product MYD021KM and the Level-2 cloud product MYD06 (Platnick et al., 2017), both with a spatial resolution of 1 km at nadir point."* (Lines 115-117)

8. Line 121: The authors state that they filter out scenes with more than 10% high clouds or ice clouds. How do the authors deal with ice/high cloud pixels in scenes where they are less than 10%? Are they set to missing values and not used in either training or classification steps?

Thank you for pointing out this oversight! We initially overlooked this detail and directly used the remaining ice/high cloud pixels in both training and classification steps, which may introduce noise into the model. In future model iterations, we will exclude these pixels by setting them to missing values, ensuring they do not interfere with our training and classification process.

9. Line 172: How is the reanalysis data co-located? Do you select the nearest timestamp or interpolated the data to MODIS observations?

We interpolated the ERA5 data temporally to match the MODIS observation time. We added a detailed explanation in the manuscript to clarify our co-location method: *"For the purpose of investigating the influence of meteorological conditions on low-cloud morphologies, we conducted some statistical analyses utilizing the co-located ERA5 reanalysis data (1° × 1°, 1-hourly) from European Centre for Medium-Range Weather Forecasts (ECMWF). The co-location is achieved by spatially selecting the nearest ERA5 grid point to each MODIS observation and temporally interpolating the ERA5 data to match the exact time of the MODIS observations. This ensures accurate alignment between the two datasets in both space and time."* (Lines 176-180)

10. A link to the classification dataset is missing in the "Data Availability" section.

The classification dataset (training, validation, and test sets) has been added to the same link as our cloud classification product dataset, along with descriptions of the variables included. We informed in the manuscript that: *"Daytime and nighttime cloud classification datasets as well as our training dataset are accessible on the [https://doi.org/10.5281/zenodo.13801408](https://doi.org/10.5281/zenodo.13801408) (Wu et al., 2024)."* (Lines 426-427)

11. No information on the file "example.xlsx" in the data repository.

Thank you for noting that. The description for the file "example.xlsx" has been added into the data repository, which is: *"File 'example.xlsx': A sample of the variable data from our cloud classification dataset, showcasing the classification results of a MODIS granule captured on January 1, 2018, at 00:25 UTC. This sample is provided to help users better understand the content of our dataset."*

**Language-related suggestions:**

Line 21: Abbreviation RFO defined in abstract is not used.

Fixed.

Line 84: dependent?

Yes, fixed.

Line 91: Prior to "Section 2 intro…", perhaps insert an introductory sentence like "The manuscript is organized as follows."

Done.

Line 184: Abbreviation ML is not defined

Fixed.

Line 210: Remove underscore after Yuan et al (2020)

Done.

Line 383: Consider changing the word "worse"

The sentence has been changed to: *"which largely explains the performance gap between our nighttime model and the daytime model proposed by Yuan et al. (2020)."* (Lines 403-404)

Line 409: "… nightly …"Do you mean nighttime?

Yes, fixed.

**Reviewer #2**

Marine low-clouds cover the majority of the ocean, and play an important role on the Earth's radiation budget. Due to a lack of local or ground-based observations, satellites become powerful tools for MLC measurement, while satellite observations over nighttime are still relatively limited. Thus, this study by Wu et al. introduced a deep-learning based method for the classification of MLC and their mesoscale morphology using MODIS observations, and a global dataset is developed as well. Both all-day model and day-time model were developed and evaluated. It is interesting to find some differences on the daytime and nighttime MLC, and distinct seasonal variations were also noticed for different MLCs. The new method as well as the resulting dataset is an important addition for the community, and the paper is well organized and presented. The paper could be considered for publication after considering following suggestions.

Thank you very much for acknowledging our work and for your valuable comments and suggestions. We fully agree your concern on the construction of our dataset, the representativeness of training data, as well as the model training and validation process. In the revised version, we have further clarified the construction of our training, validation, and testing datasets, evaluated the data representativeness, and explained the reliability of our nighttime results. Please refer to the response below for further details.

1. The quality of the training and testing dataset has been essential for DL-based models, so the datasets for the training should be carefully constructed. The 2.2 Data session gave some information on the dataset, while missed some as well. For example, Figure 1 gave some examples of MLCs of different kinds, and how was the original training dataset classified? The independency of training and testing dataset is also important, so I would suggest to introduce the testing and evaluation dataset at the Data session as well.

We apologize for any confusion caused by the lack of detailed information. Our training, validation, and test datasets are all sourced from the same manually annotated dataset. While they originate from the same data pool, they have been randomly partitioned into mutually independent subsets to ensure robust evaluation and model generalization. To clarify further, we have made some modifications to the original text, and it currently reads like: *"A total of 38,756 labeled daytime scenes were obtained, including 3,548 scenes of solid stratus, 6,277 of closed MCC, 3,345 of open MCC, 6,739 of disorganized MCC, 8,947 of clustered Cu and 9,900 of suppressed Cu. These scenes were then randomly partitioned into three mutually independent datasets for training, validation, and testing, with a distribution ratio of 3:1:1 respectively."* (Lines 129-132)

2. Cloudy and atmospheric properties show clear seasonal variations. For example, surface and atmospheric temperatures may significantly different from season to season, and this is also true for clouds. It is mentioned that only the results over the first half of 2014 were used for data training. Would such choice of results from half a year influence the DL performance?

Your suggestions are highly valuable. Indeed, the formation and development of clouds are significantly influenced by meteorological conditions, such as sea surface temperature and lower tropospheric stability (LTS), which differ from season to season. Nevertheless, as the ultimate manifestation of meteorological conditions, cloud patterns exhibit certain similarities across different regions and seasons. That is, the cloud patterns in a specific region resemble those on a global scale, and the cloud patterns in the first half of the year are similar to those throughout the entire year. Therefore, the cloud patterns contained in our dataset can largely represent the all-year and global clouds.

To validate our hypothesis, we examined the differences in the probability density functions (PDFs) of thermal radiance, cloud optical thickness (COT), and cloud morphology between our training dataset and the global full-year dataset. As shown in the Figure R4. The results revealed a substantial overlap between the two PDFs, suggesting that the training data we selected is relatively representative and can be used to substitute the global full-year dataset. Therefore, we have added a statement to the article: "*The representativeness of this dataset was validated as the probability distribution functions (PDFs) of thermal radiance data and cloud optical thickness show large overlap with those of the global and full-year dataset (Fig. S1).*" (Lines 120-122)

Despite this, in the future, we will also re-label the global dataset for all seasons, both day and night, and update our model and products in subsequent iterations.

[Figure]

Figure R4: The probability density functions (PDFs) of thermal radiance, cloud optical thickness (COT), and cloud morphology in our training dataset and the global full-year dataset.

3. Would it be possible to include the exact variables of input for different models in the flowchart of figure 2? This would be very helpful to better understood the details of the model efficiently.

Thank you for your valuable suggestion. We have included the exact input variables

in Figure 2a, as also shown in the following Figure R5, to enhance the reader's understanding of the model details.

**(a)**

[Figure]

**Figure R5.** The revised ResNet-50 model structure.

4. The example of solid stratus show relatively regular linear structure, and are such structures natural? Please double check.

As mentioned in the responds to Comments #5 from Reviewer #1, these linear structures are strip noise caused by the components of satellite sensor. This type of noise is present in the classification, but it can be identified and filtered out by the model, so it does not have a significant impact on the classification process. We have attempted conventional elimination methods, but these approaches have proven unsuccessful so far. AI-based removal techniques appear promising, but additional time is required to fully master it.

5. The training model based on daytime results is extend to nighttime observations. This is essential for the work, and could be tricky. The validation of the model for nighttime observations is very important, while only some examples were shown in Figure 4. Would it be possible to improve the validation to ensure the reliability of the results for nighttime?

As we responded to the Comments #1 from Reviewer #1, the difficulty in nighttime predictions arises from the lack of cloud thickness information. However, the COT retrieval method by Wang et al. (2022) has been proven reliable and can effectively replace MODIS COT for accurate nighttime classification. Moreover, since cloud classification requires pattern recognition rather than quantitative values, the differences in infrared radiance and COT between day and night do not significantly affect the classification.

In addition, in our response to the Major comments #1 from Reviewer #3, we examined the probability distribution functions (PDFs) of COT and thermal radiance

between our training dataset (daytime) and nighttime dataset (Figure R7), which present large overlap. The significant similarity between daytime and nighttime input data indicates less extrapolation by the model and ensures the reliability of our nighttime results.

6. Figures 7 indicates clear day and time differences between RFO of different MLCs. Could the authors give some discussions on the reasons for the differences?

Thank you for pointing us in this direction. Your suggestion actually aligns with our own thoughts, and it is part of our next steps.

In this study, we have conducted a statistical analysis of six meteorological conditions in the article but found that the variations in these meteorological factors between day and night were not significant. Therefore, other factors, such as cloud-top radiative cooling, might be responsible for the observed differences. We plan to further investigate the controlling factors behind the day-night changes of cloud morphology in future work.

Furthermore, given that the primary focus of this article is to introduce the cloud dataset and the machine learning method, we feel that including an analysis of cloud-controlling factors might shift the focus away from the main theme.

**Reviewer #3**

Review of "A Global Classification Dataset of Daytime and Nighttime Marine Low-cloud Mesoscale Morphology Based on Deep Learning Methods" by Wu et al [MS number: essd-2024-536]

This study produces a global dataset of daytime and nighttime low-cloud mesoscale morphologies (categorized into six types) using a convolutional neural network through a combination of MODIS infrared radiance data and machine-learning-retrieved cloud optical thickness. Leveraging this novel dataset, the authors analyzed the day-night contrast in climatology, seasonal cycles, and cloud properties of cloud morphologies. One of notable findings is the significant diurnal variation in the occurrence frequency of closed MCC and suppressed Cu. The primary contribution of this work lies in the generation of nighttime low-cloud morphology data, which complements the well-established daytime morphology datasets from prior studies. This advancement would inspire and enable more downstream research like understanding the diurnal cycle of cloud morphology and cloud-longwave-radiation-climate feedback. The manuscript is overall well-written and well-organized, with nice presentation of figures. However, my major concern pertains to the limitations in the model's training and validation processes, which could impact the dataset's reliability. Addressing these issues would significantly strengthen the study's contribution to the marine low cloud research community. I'd like to recommend a major revision before this manuscript is considered for publication in ESSD.

Thank you for your valuable comments and suggestions. We are pleased that the value of our nighttime dataset has been recognized. As you mentioned, this dataset can be used for various follow-up studies, such as cloud morphology diurnal cycles, the controlling factors of cloud type transitions, and cloud-longwave-radiation-climate feedbacks. Beyond these, we can also investigate the impact of climate change and anthropogenic emissions reductions on the long-term trends of cloud morphology, providing a more comprehensive evaluation of cloud radiative feedbacks associated with type transitions. We hope this research will encourage greater attention to the indirect climate effects of marine low-cloud morphology, which is also the primary purpose of our dataset: to be widely used by the community and to offer valuable insights.

Regarding the limitations you pointed in the model's training and validation process, we fully agree with your concerns. In the revised manuscript, we have further clarified the model's training and validation processes, validated the global applicability of the regionally trained model, and fixed some minor issues. Please see the response below for further details.

At last, your thoughts actually align closely with ours. Many of the suggestions you mentioned, such as investigating the physical reasons behind the diurnal variation of cloud morphology and the impact of morphology transitions on shortwave and longwave radiation at the TOA, are already ongoing within our group. However, since this article is more foundational, many aspects were not fully presented. We look forward to sharing new results with you soon.

**Major comments:**

1. One of my primary concerns is the validity of applying a regionally trained deep learning (DL) model to global predictions. In this study, the authors developed their model using data from the SEP region only and then applied it to generate a global dataset. While the model demonstrates relatively high prediction accuracy over SEP (Figure 3), it is unclear whether this performance extends to global applications. Regarding this issue, the authors should first clarify the rationale for selecting SEP as the training region rather than using a global or other regional dataset. Was this choice subject to the limited availability of the data, or is there a similarity in morphology climatology between SEP and the global scale? If SEP is your best choice at the moment, it would be essential to evaluate whether using a regionally trained model for global predictions is reasonable. One approach to examine this would be to generate a global map of prediction accuracy for each cloud morphology type to check the model's global performance. Additionally, the authors could examine the differences in the PDFs of thermal radiance, COT, and cloud morphology between SEP and the global dataset. A smaller difference or larger overlaps would indicate less extrapolation by the model, enhancing the credibility of the global dataset.

Similarly, the authors would have to be careful when extending the daytime-trained model to nighttime predictions, as this may also introduce potential extrapolation issues. The authors provided only a single example to illustrate the model's success at nighttime, which is insufficient to establish its statistical reliability. To address this concern, additional cases should be analyzed to validate the model's nighttime performance. Alternatively, examining the differences in the PDFs of thermal radiance, COT, and cloud morphology between daytime and nighttime could help assess the extent of extrapolation and ensure the robustness of the predictions.

We totally agree with your ideas. Applying a model trained on regional datasets to global dataset need careful validation.

First, we give the rationale for selecting the SEP region as the training dataset:

1) We believe that SEP region encompass all the cloud types around the world, and can provide sufficient samples for each type.

2) Meanwhile, we are limited by the available training data. We only labeled this portion of the data initially, and re-labeling the global dataset and regenerating the cloud dataset would take a considerable amount of time.

However, your suggestions provided effective methods for our model validation. Following your suggestion, we generated the plots of probability distribution functions (PDFs) of COT, thermal radiance, and cloud morphology. One plot compares our training dataset with the global full-year dataset (Figure R6), while the other compares the nighttime dataset (Figure R7).

Both figures show large overlap in the PDFs of COT and thermal radiance. So, we have added some statements in the manuscript to validate the reliability of our model and the nighttime results: *"The representativeness of this dataset was validated as the probability distribution functions (PDFs) of thermal radiance data and cloud optical thickness show large overlap with those of the global and full-year dataset (Fig. S1)"* (Lines 120-122)

*"In addition, we further examined the differences in the PDFs of the thermal radiance data and the TIR-CNN-retrieved COT between daytime and nighttime. As depicted in Fig. S4, these PDFs nearly overlapped, which means less extrapolation will be introduced when the model is generalized to nighttime data. And it also illustrates the credibility of our nighttime classification dataset."* (Lines 145-148)

In addition, due to the manual addition and reduction of sample sizes for each cloud type in our training dataset, the cloud morphology PDFs differ slightly from the other two datasets. However, this was a deliberate choice made to improve our model performance.

[Figure]

**Figure R6.** The probability distribution functions (PDFs) of COT, thermal radiance, and cloud morphology between our training dataset and the global full-year dataset.

[Figure]

**Figure R7.** The probability distribution functions (PDFs) of COT, thermal radiance, and cloud morphology between our training dataset (daytime) and nighttime dataset.

2. Regarding the model training, validation, and testing, the data-splitting strategy is unclear. For instance, was the dataset split randomly or manually into the 6:2:2 ratio? Furthermore, the validation method used to assess the model's predictions has not been described. The authors should clarify these aspects to improve the robustness of their results.

We fully agree that clarifying our dataset splitting method and model validation method is necessary. Indeed, the original annotated dataset was randomly split according to a 3:1:1 ratio, so we clarified it in the manuscript: *"These scenes were then randomly partitioned into three mutually independent datasets for training, validation, and testing, with a distribution ratio of 3:1:1 respectively."* (Lines 131-132)

Based on your suggestions, we have added the following content regarding to the validation method in the "2.4 Method" Section: *"After each training epoch, the validation dataset was used to evaluate the trained model's performance, which allows us to monitor the model's success. When the training process was completed, the test dataset was used for the final evaluation of the model's performance. We used the optimal model to predict each sample in the test dataset, compared the model's predictions with the true labels, and assessed the accuracy using metrics such as accuracy, F1-score, and recall."* (Lines 229-232)

And we have also added the content regarding model test results in Section '3.1 Model Performance': *"Our model achieves an average precision of approximately 91%, an F1-score of 90.6%, and a recall of 90.8%, demonstrating its strong generalization capability and robustness."* (Lines 241-243)

3. Given the critical role of cloud morphologies in Earth's radiation budget, the authors could consider including a climatological analysis of shortwave and longwave radiation at the TOA for the six cloud morphology types. Adding such an analysis would significantly enhance the insights and scientific value of this study.

Thank you for your insightful and constructive suggestions! Your idea aligns perfectly with our thoughts. We are currently conducting a long-term trend study on cloud morphology using this cloud dataset, as well as exploring the impact of aerosols and climate change on these long-term trends. We aim to conduct a systematic study of cloud morphology and cloud type transitions. Based on this, our next step will be to investigate radiative effects. However, some issues are existing in the current radiative datasets: there is a lack of instantaneous clear sky albedo. We need to address some fundamental issues to make the radiative data more solid, which will take time. Moreover, since our cloud scenes are instantaneous rather than monthly, matching the five years of radiative data will be slow, and this could eventually become a separate paper.

In earlier research, Mohrmann et al. (2021) have assessed the radiative properties of the six cloud types, using data from the Clouds and the Earth's Radiant Energy System (CERES), specifically SYN 1-degree hourly data (daytime). They analyzed the net cloud radiative effect (CRE) for each cloud type at different spatial scales and found that Solid MCC and Disorganized MCC exhibit the strongest climatological average cloud radiative effects (Figure R8c, square symbol, about -48 W m$^{-2}$). However, they did not account for global radiative impacts associated with long-term changes in cloud morphology, which is the question we aim to address in future work.

[Figure]

**Figure R8.** Cloud radiative properties by cloud type from Mohrmann et al. (2021): (a) CERES cloud fraction, (b) cloud frequency of occurrence, (c) average CERES net CRE per cloud type, (d) frequency-weighted net CRE. Each set of three symbols is for the 3 years (2014-2016) used. For panels (a) and (c), the mesoscale, synoptic, climatological averages are shown using circular, diamond, and square symbols, respectively.

**Minor comments:**

L30: longwave warming effects are more significant for high clouds, which might not be so for low clouds.

The original sentence was changed to: *"They exert a strong radiative cooling on the planet as the residual of a larger cooling effect and a positive warming effect (Klein and Hartmann, 1993; Eytan et al., 2020)."* (Lines 30-32)

L67: What is the major difference between the six-type classification of this study and the four type one here?

The major differences are: (1) Scale: The four types here require a larger scale to be fully observed, at least a 10° by 10° field of view. (2) Classification Basis: Our

six-type classification emphasizes the underlying physical processes, while their four-type classification focuses primarily on the external appearance and morphological features.

We have incorporated some key information into the original sentence, and it now reads as follows: *"Moreover, Schulz et al. (2021) developed an object detection model to classify four larger scale ($10^o \times 10^o$) cloud morphologies in trade wind regions of North Atlantic. These morphologies were vividly named as "sugar," "gravel," "flowers," and "fish" mainly based on their visual appearances." (Lines 66-69)*

L81: Do you mean the decline in the *long-term* trend?

Yes, fixed.

L83: "how much they contribute to … remain unclear" to "how nighttime cloud cover varies under different cloud morphology types remain unclear."

Done.

L90: Please clarify the temporal and spatial resolution.

Done.

L97: "created" to "driven"

Done.

L119: Please clarify the temporal resolution of the training dataset.

Done.

L121-122: Have you excluded middle clouds (i.e., those situated between 3 and 6 km)? These clouds are prevalent over midlatitude oceans, and they also contaminate low cloud observations.

We are sorry for not eliminating the influence of middle clouds in our training process. In future work, we plan to refine our methodology by re-screening middle clouds to improve the accuracy of our model.

L174: Please clarify the level of the divergence used.

Done.

L199: I'd suggest labeling the input variables (three channels and COT) and the output variables (six cloud morphology types) in Figure 2a to improve its clarity and readability.

Done.

L210: It looks like the improvement is limited. Have you examined the COT retrieval uncertainty? If it is greater than the improved accuracy, it would be unnecessary to include the COT into the predictors.

Thank you for your suggestion. As mentioned in Comments #1 from Reviewer #1, the model using retrieved COT demonstrates comparable predictive accuracy to the model based on MODIS COT, confirming its reliability.

L210: Typo: "Yuan et al (2020)_due to" to "Yuan et al (2020) due to"

Fixed.

L212: Which is it relative to?

We have completed the sentence as follows: *" Although it is a bit less accurate compared to the visible light model from Yuan et al. (2020), it is undeniable that this model has achieved a relatively high accuracy level when compared to other TIR*

*model (Lang et al., 2022), and can effectively accomplish the classification tasks we proposed.”* (Lines 237-240)

L219: "clustered Cu" to "clustered Cu or closed MCC"

Done.

L250: "n denotes" to "with n denoting"

Done.

L305: "its seasonal variation" to "the peak in summer"

Done.

L332: do you mean "decrease by 2 microns on average"?

Yes, fixed.

L333: Please clarify whether the LWP mentioned here represents the in-cloud value or the grid box mean value.

Sorry for the lack of information provided. We have added the following clarification in Lines 357-358: *"All of the cloud microphysical properties represent the in-cloud mean value within a 1° grid."*

L349: Why is there a westward shift at night? Also, for stratocumulus clouds, LTS is usually higher at night. Why does it decline for closed MCC at night?

Sorry for the confusion. We originally hypothesize that the observed decline in LTS at night may be related to the movement of clouds. Since our statistical analysis covers a relatively large spatial scale, the meteorological conditions associated with specific cloud types could change as the clouds move, potentially leading to the observed decrease in LTS. However, we apologize for the previous implication that this movement is necessarily westward. So, we removed the mention of "westward" in the revised text. We believe that further investigation is required to fully understand the reasons behind the LTS decline in closed MCCs at night.

L351: It would be more interesting to discuss their physical reason.

Thank you for pointing the direction for us, it is the next part of our work. It is difficult to determine the key cloud-controlling factors for each cloud type based solely on the statistical analysis in this study, and many other environmental conditions have not been included. Therefore, we plan to conduct a more detailed and comprehensive investigation into their physical causes in our next work. Furthermore, since the main focus of this article is to introduce a cloud dataset, we feel that including an analysis of cloud-controlling factors here would dilute the main theme.

L367: Why are the results shown here only for SEP, while Figure 10 presents global results?

Cloud morphology is controlled by multiple meteorological factors (Liu et al., 2024). When we study the influence of one controlling factor, cloud morphologies can be affected by the variability in other factors if the study is conducted in a global scale. For instance, while exist within the same LTS environment, clouds in mid-to-high latitude regions and those near the equator have distinctly different sea surface temperatures. Therefore, by restricting the region, we can facilitate the day-night comparison of the primary controlling factors while excluding the interference from other variables. Clouds properties are the final result of all meteorological conditions, and their properties show little differences across different regions.

**References**

Eytan, E., Koren, I., Altaratz, O., Kostinski, A. B., and Ronen, A.: Longwave radiative effect of the cloud twilight zone, Nature Geoscience, 13, 669-673, https://doi.org/10.1038/s41561-020-0636-8, 2020.

Klein, S. A. and Hartmann, D. L.: The Seasonal Cycle of Low Stratiform Clouds, Journal of Climate, 6, 1587-1606, https://doi.org/10.1175/1520-0442(1993)006<1587:TSCOLS>2.0.CO;2, 1993.

Lang, F., Ackermann, L., Huang, Y., Truong, S. C. H., Siems, S. T., and Manton, M. J.: A climatology of open and closed mesoscale cellular convection over the Southern Ocean derived from Himawari-8 observations, Atmos. Chem. Phys., 22, 2135-2152, https://doi.org/10.5194/acp-22-2135-2022, 2022.

Liu, J., Zhu, Y., Wang, M., and Rosenfeld, D.: Cloud Susceptibility to Aerosols: Comparing Cloud-Appearance vs. Cloud-Controlling Factors Regimes, EGU General Assembly 2024, Vienna, Austria, 14–19 Apr 2024, EGU24-4059, https://doi.org/10.5194/egusphere-egu24-4059, 2024.

McCoy, I. L., McCoy, D. T., Wood, R., Zuidema, P., and Bender, F. A.-M.: The Role of Mesoscale Cloud Morphology in the Shortwave Cloud Feedback, Geophysical Research Letters, 50, e2022GL101042, https://doi.org/10.1029/2022GL101042, 2023.

Mohrmann, J., Wood, R., Yuan, T., Song, H., Eastman, R., and Oreopoulos, L.: Identifying meteorological influences on marine low-cloud mesoscale morphology using satellite classifications, Atmos. Chem. Phys., 21, 9629-9642, https://doi.org/10.5194/acp-21-9629-2021, 2021.

Platnick, S., Meyer, K. G., King, M. D., Wind, G., Amarasinghe, N., Marchant, B., Arnold, G. T., Zhang, Z., Hubanks, P. A., Holz, R. E., Yang, P., Ridgway, W. L., and Riedi, J.: The MODIS Cloud Optical and Microphysical Products: Collection 6 Updates and Examples From Terra and Aqua, IEEE Transactions on Geoscience and Remote Sensing, 55, 502-525, https://doi.org/10.1109/TGRS.2016.2610522, 2017.

Schulz, H., Eastman, R., and Stevens, B.: Characterization and Evolution of Organized Shallow Convection in the Downstream North Atlantic Trades, Journal of Geophysical Research: Atmospheres, 126, e2021JD034575, https://doi.org/10.1029/2021JD034575, 2021.

Stevens, B., Bony, S., Brogniez, H., Hentgen, L., Hohenegger, C., Kiemle, C., L'Ecuyer, T. S., Naumann, A. K., Schulz, H., Siebesma, P. A., Vial, J., Winker, D. M., and Zuidema, P.: Sugar, gravel, fish and flowers: Mesoscale cloud patterns in the trade winds, Quarterly Journal of the Royal Meteorological Society, 146, 141-152, https://doi.org/10.1002/qj.3662, 2020.

Wang, Q., Zhou, C., Zhuge, X., Liu, C., Weng, F., and Wang, M.: Retrieval of cloud properties from thermal infrared radiometry using convolutional neural network, Remote Sensing of Environment, 278, 113079, https://doi.org/10.1016/j.rse.2022.113079, 2022.

Wu, Y., Liu, J., Zhu, Y., Zhang, Y., Cao, Y., Huang, K.-E., Zheng, B., Wang, Y., Wang, Q., Zhou, C., Liang, Y., Wang, M., and Rosenfeld, D.: Global Classification Dataset of Daytime and Nighttime Marine Low-cloud Mesoscale Morphology [dataset], https://doi.org/10.5281/zenodo.13801408, 2024.

Yuan, T., Song, H., Wood, R., Mohrmann, J., Meyer, K., Oreopoulos, L., and Platnick, S.: Applying deep learning to NASA MODIS data to create a community record of marine low-cloud mesoscale morphology, Atmos. Meas. Tech., 13, 6989-6997, https://doi.org/10.5194/amt-13-6989-2020, 2020.

---

## Author Comment (AC3)

Dear Reviewer,

We are grateful to your constructive comments and valuable suggestions, which helped us to further improve our manuscript and dataset. Below we address your concerns point-by-point, with the original comments in **black** and our response in *blue*. The revised sentences in the manuscript are indicated in *italics*.

**Overview:**

Marine low-clouds cover the majority of the ocean, and play an important role on the Earth's radiation budget. Due to a lack of local or ground-based observations, satellites become powerful tools for MLC measurement, while satellite observations over nighttime are still relatively limited. Thus, this study by Wu et al. introduced a deep-learning based method for the classification of MLC and their mesoscale morphology using MODIS observations, and a global dataset is developed as well. Both all-day model and day-time model were developed and evaluated. It is interesting to find some differences on the daytime and nighttime MLC, and distinct seasonal variations were also noticed for different MLCs. The new method as well as the resulting dataset is an important addition for the community, and the paper is well organized and presented. The paper could be considered for publication after considering following suggestions.

Thank you very much for acknowledging our work and for your valuable comments and suggestions. We fully agree your concern on the construction of our dataset, the representativeness of training data, as well as the model training and validation process. In the revised version, we have further clarified the construction of our training, validation, and testing datasets, evaluated the data representativeness, and explained the reliability of our nighttime results. Please refer to the response below for further details.

1. The quality of the training and testing dataset has been essential for DL-based models, so the datasets for the training should be carefully constructed. The 2.2 Data session gave some information on the dataset, while missed some as well. For example, Figure 1 gave some examples of MLCs of different kinds, and how was the original training dataset classified? The independency of training and testing dataset is also important, so I would suggest to introduce the testing and evaluation dataset at the Data session as well.

We apologize for any confusion caused by the lack of detailed information. Our training, validation, and test datasets are all sourced from the same manually annotated dataset. While they originate from the same data pool, they have been randomly partitioned into mutually independent subsets to ensure robust evaluation and model generalization. To clarify further, we have made some modifications to the original text, and it currently reads like: *"A total of 38,756 labeled daytime scenes were obtained, including 3,548 scenes of solid stratus, 6,277 of closed MCC, 3,345 of*

*open MCC, 6,739 of disorganized MCC, 8,947 of clustered Cu and 9,900 of suppressed Cu. These scenes were then randomly partitioned into three mutually independent datasets for training, validation, and testing, with a distribution ratio of 3:1:1 respectively."* (Lines 129-132)

2. Cloudy and atmospheric properties show clear seasonal variations. For example, surface and atmospheric temperatures may significantly different from season to season, and this is also true for clouds. It is mentioned that only the results over the first half of 2014 were used for data training. Would such choice of results from half a year influence the DL performance?

Your suggestions are highly valuable. Indeed, the formation and development of clouds are significantly influenced by meteorological conditions, such as sea surface temperature and lower tropospheric stability (LTS), which differ from season to season. Nevertheless, as the ultimate manifestation of meteorological conditions, cloud patterns exhibit certain similarities across different regions and seasons. That is, the cloud patterns in a specific region resemble those on a global scale, and the cloud patterns in the first half of the year are similar to those throughout the entire year. Therefore, the cloud patterns contained in our dataset can largely represent the all-year and global clouds.

To validate our hypothesis, we examined the differences in the probability density functions (PDFs) of thermal radiance, cloud optical thickness (COT), and cloud morphology between our training dataset and a global full-year dataset. As shown in the Figure R1. The results revealed a substantial overlap between the two PDFs, suggesting that the training data we selected is relatively representative and can be used to substitute the global full-year dataset. Therefore, we have added a statement to the article: *"The representativeness of this dataset was validated as the probability density functions (PDFs) of thermal radiance data and cloud optical thickness show large overlap with those of the global and full-year dataset (Fig. S1)."* (Lines 120-122)

Despite this, in the future, we plan to re-label the global dataset for all seasons, both day and night, and update our model and products in subsequent iterations.

[Figure]

**Figure R1.** The comparison of probability density functions (PDFs) between our training dataset and a global full-year dataset. (a) PDFs of cloud optical thickness

(COT); PDFs of radiance data from infrared channels: (b) 29, (c) 31, (d) 32; (e) PDFs of cloud morphology.

3. Would it be possible to include the exact variables of input for different models in the flowchart of figure 2? This would be very helpful to better understood the details of the model efficiently.

Thank you for your valuable suggestion. We have included the exact input variables in Figure 2a, as also shown in the following Figure R2, to enhance the reader's understanding of the model details.

[Figure]

**Figure R2.** The revised ResNet-50 model structure figure.

4. The example of solid stratus show relatively regular linear structure, and are such structures natural? Please double check.

These linear structures are strip noise caused by the components of satellite sensor. Although present in the classification processes, they can be identified and filtered out by the model, thus having minimal impact on the training and classification results. We have attempted conventional methods, including mean filtering, Fourier transform, and directional filtering, to eliminate these strips and enhance the visual quality; however, none of them have proven effective yet (Figure R3). AI-based removal techniques appear promising, but additional time is required to fully master it.

[Figure]

**Figure R3.** The images processed with directional filtering.

5. The training model based on daytime results is extend to nighttime observations. This is essential for the work, and could be tricky. The validation of the model for nighttime observations is very important, while only some examples were shown in Figure 4. Would it be possible to improve the validation to ensure the reliability of the results for nighttime?

We agree with your concern. The difficulty in nighttime predictions arises from the lack of cloud thickness information. However, as we responded to the Comments #1 from Reviewer #1, the COT retrieval method by Wang et al. (2022) has been proven reliable and can effectively replace MODIS COT for accurate nighttime classification (Figure R4). Moreover, since cloud classification requires pattern recognition rather than quantitative values, the differences in infrared radiance and COT value between day and night do not significantly affect the classification.

In addition, we examined the probability density functions (PDFs) of COT and thermal radiance between our training dataset (daytime) and a nighttime dataset (Figure R5), which present large overlap. The significant similarity between daytime and nighttime input data indicates less extrapolation by the model and ensures the reliability of our nighttime results. Therefore, we added some statements in the manuscript: *"In addition, we further examined the differences in the PDFs of the thermal radiance data and the TIR-CNN-based COT between our training dataset (daytime) and nighttime dataset. As depicted in Fig. S4, these PDFs nearly overlapped, which means less extrapolation will be introduced when the model is generalized to nighttime data. And it also illustrates the credibility of our nighttime classification results."* (Lines 145-148)

[Figure]

**Figure R4.** Model training results based on MODIS COT. (a) the model's accuracy on the training and validation datasets. (b) the confusion matrix of the model.

[Figure]

**Figure R5.** The comparison of probability density functions (PDFs) between our training dataset and a nighttime dataset. (a) PDFs of cloud optical thickness (COT); PDFs of radiance data from infrared channels: (b) 29, (c) 31, (d) 32; (e) PDFs of cloud morphology.

6. Figures 7 indicates clear day and time differences between RFO of different MLCs. Could the authors give some discussions on the reasons for the differences?

Thank you for pointing us in this direction. Your suggestion actually aligns with our own thoughts, and it is part of our next steps.

In this study, we have conducted a statistical analysis of six meteorological conditions in the article but found that the variations in these meteorological factors between day and night were not significant. Therefore, other factors, such as cloud-top radiative cooling, might be responsible for the observed differences. We plan to further investigate the controlling factors behind the day-night changes of cloud morphology in future work.

Furthermore, given that the primary focus of this article is to introduce the cloud dataset and the machine learning method, we feel that including an analysis of cloud-controlling factors might shift the focus away from the main theme.

**References**

Wang, Q., Zhou, C., Zhuge, X., Liu, C., Weng, F., and Wang, M.: Retrieval of cloud properties from thermal infrared radiometry using convolutional neural network, Remote Sensing of Environment, 278, 113079, https://doi.org/10.1016/j.rse.2022.113079, 2022.